# In vivo single-molecule analysis reveals *COOLAIR* RNA structural diversity

Minglei Yang[1,2], Pan Zhu[1,2], Jitender Cheema[1], Rebecca Bloomer[1], Pawel Mikulski[1], Qi Liu[1], Yueying Zhang[1], Caroline Dean[1✉] & Yiliang Ding[1✉]

Cellular RNAs are heterogeneous with respect to their alternative processing and secondary structures, but the functional importance of this complexity is still poorly understood. A set of alternatively processed antisense non-coding transcripts, which are collectively called *COOLAIR*, are generated at the *Arabidopsis* floral-repressor locus *FLOWERING LOCUS C* (*FLC*)[1]. Different isoforms of *COOLAIR* influence *FLC* transcriptional output in warm and cold conditions[2–7]. Here, to further investigate the function of *COOLAIR*, we developed an RNA structure-profiling method to determine the in vivo structure of single RNA molecules rather than the RNA population average. This revealed that individual isoforms of the *COOLAIR* transcript adopt multiple structures with different conformational dynamics. The major distally polyadenylated *COOLAIR* isoform in warm conditions adopts three predominant structural conformations, the proportions and conformations of which change after cold exposure. An alternatively spliced, strongly cold-upregulated distal *COOLAIR* isoform[6] shows high structural diversity, in contrast to proximally polyadenylated *COOLAIR*. A hyper-variable *COOLAIR* structural element was identified that was complementary to the *FLC* transcription start site. Mutations altering the structure of this region changed *FLC* expression and flowering time, consistent with an important regulatory role of the *COOLAIR* structure in *FLC* transcription. Our work demonstrates that isoforms of non-coding RNA transcripts adopt multiple distinct and functionally relevant structural conformations, which change in abundance and shape in response to external conditions.

*COOLAIR* transcripts are alternatively polyadenylated at proximal sites to give around 400-nucleotide (nt) class I transcripts, or at distal sites to give around 600–750-nt class II transcripts[1] (Fig. 1a). The different *COOLAIR* isoforms have been functionally linked to R-loop-mediated chromatin silencing, transcriptional derepression in warm-grown plants[2,7] and *FLC* transcriptional silencing in the cold[3,4,6], through as yet poorly understood mechanisms. The secondary structure of RNA is emerging as an important regulator of RNA function[8]. Structural analysis of in vitro synthesized *COOLAIR* revealed the evolutionary conservation of class II *COOLAIR* structures, despite low nucleotide sequence identity[5]. However, knowledge of the *COOLAIR* structure in vivo is necessary to understand the function and complexity of *COOLAIR* in living cells. Current chemical probing methods were limiting for this purpose for two reasons: first, it has not been possible to accurately profile the full-length structural landscape and distinguish structures in shared regions between isoforms using short-read sequencing platforms; second, RNA conformational heterogeneity complicates querying the RNA secondary structures after chemical probing. Despite recent improvements in these techniques[9–11] (Supplementary Discussion), the ability to directly identify different RNA isoforms and determine single-molecule in vivo conformations was still difficult. We therefore developed a single-molecule-based RNA secondary structure probing

method that enables the direct determination of structural conformations of individual RNA isoforms.

## Structural diversity of *COOLAIR* isoforms

*COOLAIR* is involved both in modulating the *FLC* transcriptional output to determine the winter annual or rapid-cycling reproductive strategy of warm-grown plants[2] and in facilitating the cold-induced transcriptional shut-down that precedes stable epigenetic silencing of Polycomb Repressive Complex 2 in vernalization[3,4,6]. We therefore profiled the in vivo RNA secondary structure landscapes of all of the major isoforms, that is, class I and class II *COOLAIR* transcript isoforms (Fig. 1a and Extended Data Fig. 1a) in wild-type plants (Col *FRI*) grown in warm conditions and after two weeks of cold exposure when *FLC* is transcriptionally downregulated[1,12]. RNA structure determination was carried out using in vivo selective 2′-hydroxyl acylation analysed by primer extension (SHAPE) chemical probing in *Arabidopsis thaliana* seedlings. The SHAPE reagent, 2-methylnicotinic acid imidazolide (NAI), modifies single-stranded sites of all four RNA nucleotides[13]. The extracted RNAs were reverse transcribed, and the modified sites led to mutations in the complementary DNA (cDNA) (Fig. 1a). We then adapted the resulting cDNAs into the PacBio

[1]John Innes Centre, Norwich Research Park, Norwich, UK. [2]These authors contributed equally: Minglei Yang, Pan Zhu. ✉e-mail: caroline.dean@jic.ac.uk; yiliang.ding@jic.ac.uk

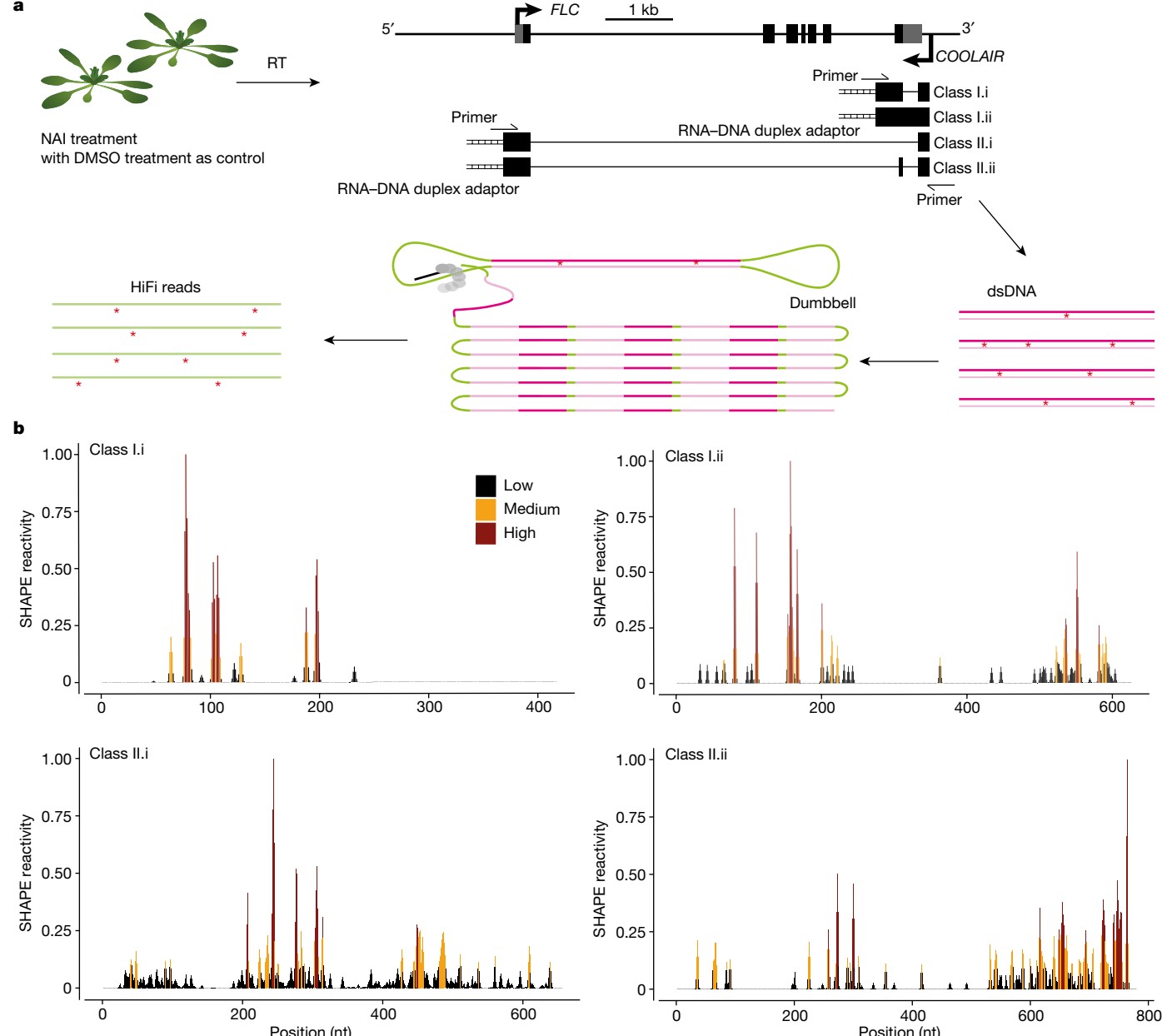

**Fig. 1 | smStructure-seq captures RNA secondary structure information of different transcript isoforms. a**, Schematic of the smStructure-seq design for RNA secondary structure probing of each *COOLAIR* isoform. The *Arabidopsis* seedlings were treated with NAI ((+)SHAPE) or DMSO ((−)SHAPE). Total RNA was extracted, and the RNA–DNA hybrid adaptors (ladder symbol) were added to the reverse-transcription (RT) reaction using TGIRT-III enzyme. dsDNAs were generated by adding specific primers for all of the *COOLAIR* isoforms. The dumbbell adaptors were then ligated to the resulting dsDNAs to generate PacBio libraries. The raw subreads were converted to high-accuracy HiFi reads (or circular consensus sequences)[14] to generate the mutation rate profiles. **b**, The normalized SHAPE reactivities derived from the mutation rate profiles were plotted for different class I (under cold-grown conditions) and II (under warm-grown conditions) *COOLAIR* transcript isoforms. The normalized SHAPE reactivity is calculated from merged *n* = 2 biological replicates. These reactivity values are colour-coded and shown on the *y* axis.

platform for single-molecule real-time sequencing, which we call single-molecule-based RNA structure sequencing (smStructure-seq). The derived raw reads were processed to obtain high-accuracy HiFi reads[14] to generate the SHAPE reactivities based on the NAI-adduct mutational profiles (Fig. 1a). To benchmark the reproducibility and accuracy of our smStructure-seq data, we calculated the SHAPE reactivities of 18S rRNA. We found that our smStructure-seq libraries were highly reproducible with very high Pearson correlations of 0.95 (*P* value = 0.2 × 10$^{-16}$). By comparing our SHAPE reactivities with the 18S rRNA phylogenetic secondary structure[15], we found that our smStructure-seq analysis can accurately investigate the full-length

RNA structure in vivo (a detailed explanation is provided in the legend of Extended Data Fig. 1b).

We next directly calculated the SHAPE reactivity profiles for class I.i, class I.ii, class II.i and class II.ii *COOLAIR* isoforms in warm and cold conditions (Fig. 1b and Extended Data Fig. 1c). Class I.i and class I.ii showed relatively few nucleotides with SHAPE reactivity (more than 95% nucleotides of class I isoforms showed no NAI-adduct mutation in warm-grown plants) (Extended Data Fig. 1c). The *COOLAIR* class I transcripts are associated with a stable R-loop structure[2], potentially accounting for this low reactivity. In the same sample, the SHAPE reactivities of class II isoforms in warm-grown plants were much higher

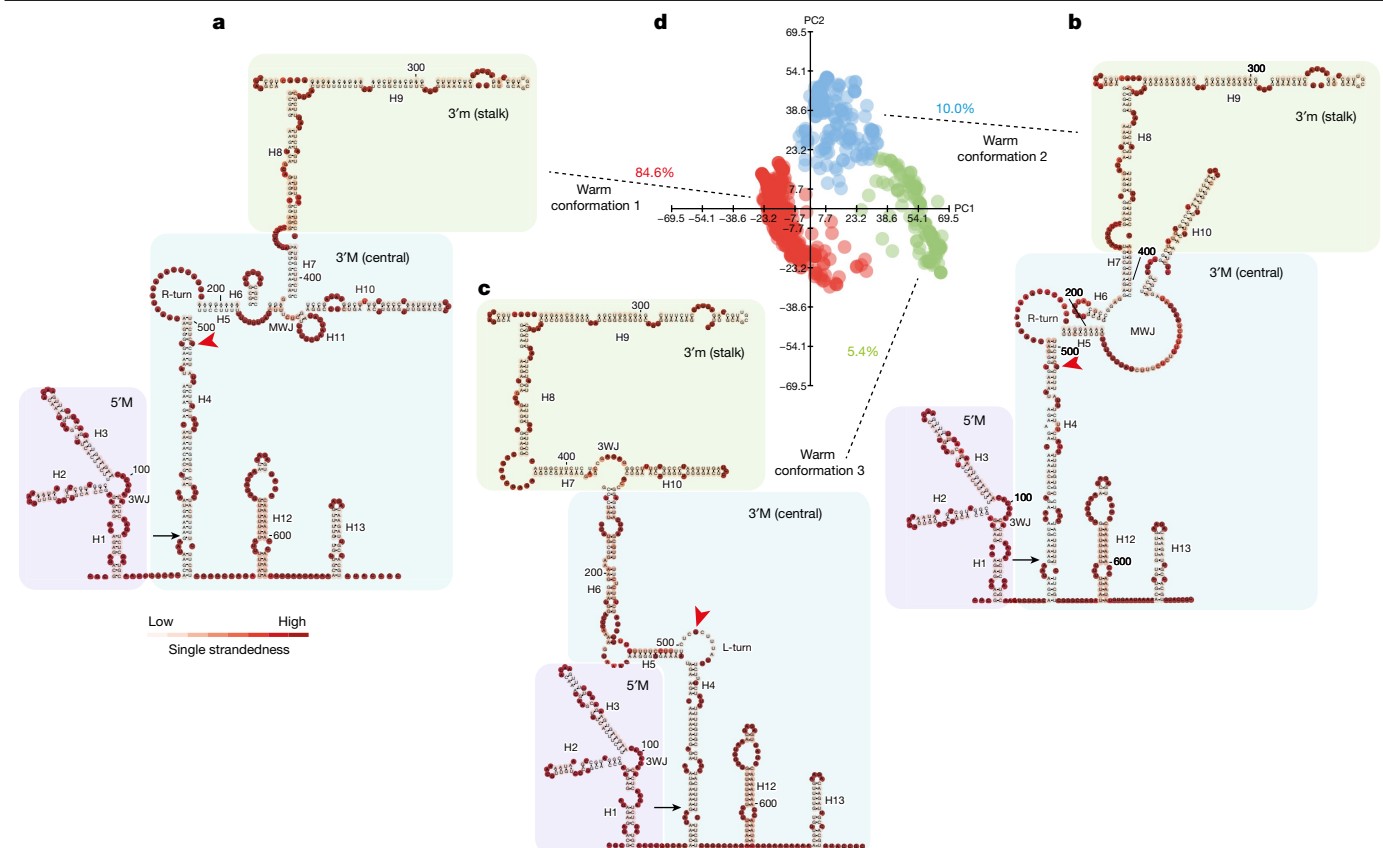

**Fig. 2 | The three major conformations of class II.i in warm-grown plants. a–c**, Representative structural models of warm conformation 1 (**a**), warm conformation 2 (**b**) and warm conformation 3 (**c**) from **d**. Models were coloured according to the likelihood of single strandedness. The red arrowheads indicate the site corresponding to the *FLC* TSS. **d**, Visualization of the in vivo structural conformations of class II.i in warm-grown plants. Structures were directly generated from 3,061 individual mutational profiles. Data were visualized using principal component analysis (PCA). Each dot represents a unique single structure derived from each single-molecule mutational profile. 3WJ, three-way junction; H#, helix number; MWJ, multiway junction; L-turn, left-handed turn motif; PC, principal component; R-turn, right-handed turn motif. Black arrow, *COOLAIR* exon boundary; red arrowhead, reverse-complementary to the *FLC* TSS.

(Fig. 1b and Extended Data Fig. 1c). The overall SHAPE profiles were notably different between class II.i and class II.ii (Fig. 1b), even though most of these two isoforms were composed of the same sequence.

Thermodynamic parameter-based RNA structure analysis aims to find the thermodynamically favourable RNA structure[16]. However, long noncoding RNAs (lncRNAs), such as *COOLAIR*, are dynamically involved in co-transcriptional regulation and, therefore, thermodynamics may have an incomplete role in determining the RNA structure in vivo[17]. We therefore developed an analysis method for our smStructure-seq that adopted stochastic context-free grammar (SCFG) constrained by individual SHAPE reactivity profiles, enabling the determination of the RNA structure of single-RNA molecules independent of thermodynamics. We named this structural analysis method DaVinci (Determination of the Variation of the RNA structure conformation through stochastic context-free grammar). DaVinci can construct a wide RNA structure landscape by generating the conformation of individual RNA structures from each in vivo SHAPE mutational profile (Extended Data Fig. 2a). Because DaVinci takes advantage of each single mutational profile rather than the averaged SHAPE mutational profiles, it can identify each possible conformation at single-molecule resolution. To exemplify this, we found that DaVinci could identify a cryptic conformation (conformation 3) of the HIV Rev response element (RRE)[18] that was not identified by the chemical-reactivity-based clustering method[11] (Extended Data Fig. 2b–e). This cryptic conformation becomes the major conformation when introducing mutations in RRE61 (Supplementary Discussion; more validations are shown in Extended Data Figs. 2f–h and 3). Using

DaVinci, we identified at least three major structural conformations of *COOLAIR* class II.i, the most abundant (Extended Data Fig. 1a) class II isoform in warm conditions (84.6% warm conformation 1; 10% warm conformation 2 and 5.4% warm conformation 3; Fig. 2a–d). These in vivo structural conformations are organized into three domains (Fig. 2a–c): the 5′ domain in exon 1; the 3′ major domain (3′M) or central domain in exon 2; and the 3′ minor domain (3′m), stalk domain also in exon 2. All three warm conformations show a certain similarity to the in vitro class II.i structure[5], in the 5′ domain and the 3′m domains, but are distinct in the central 3′M domain (Extended Data Fig. 4a,c,d). Consistently, both measurements of topological similarity (tree alignment, TA) and base-pairing similarity (positive predictive value, PPV) showed that most differences between the in vitro structure and the conformations in the warm conditions are in the central domain (3′M domain) (Extended Data Fig. 4a–d). Notably, this region was proposed to be changed by a single natural nucleotide polymorphism in *A. thaliana* accession Var2–6 (ref. [7]), which enhances the production of class II.iv (Extended Data Fig. 5a), a very rare transcript in Col *FRI*[7]. Class II.iv increases *FLC* expression through a co-transcriptional mechanism that involves the capping of the *FLC* nascent transcript[7]. We performed smStructure-seq on a genotype that carries the Var2–6 *FLC* allele introgressed into Col *FRI* (Extended Data Fig. 5b). The in vivo structure of class II.iv has a very short helix 4 (H4) and a merged H5 to extend H6 (Extended Data Fig. 5b,c). These structural changes occur in the region complementary to the *FLC* transcription start site (TSS) (Extended Data Fig. 5b,c). Thus, the greatest conformational variation in distally

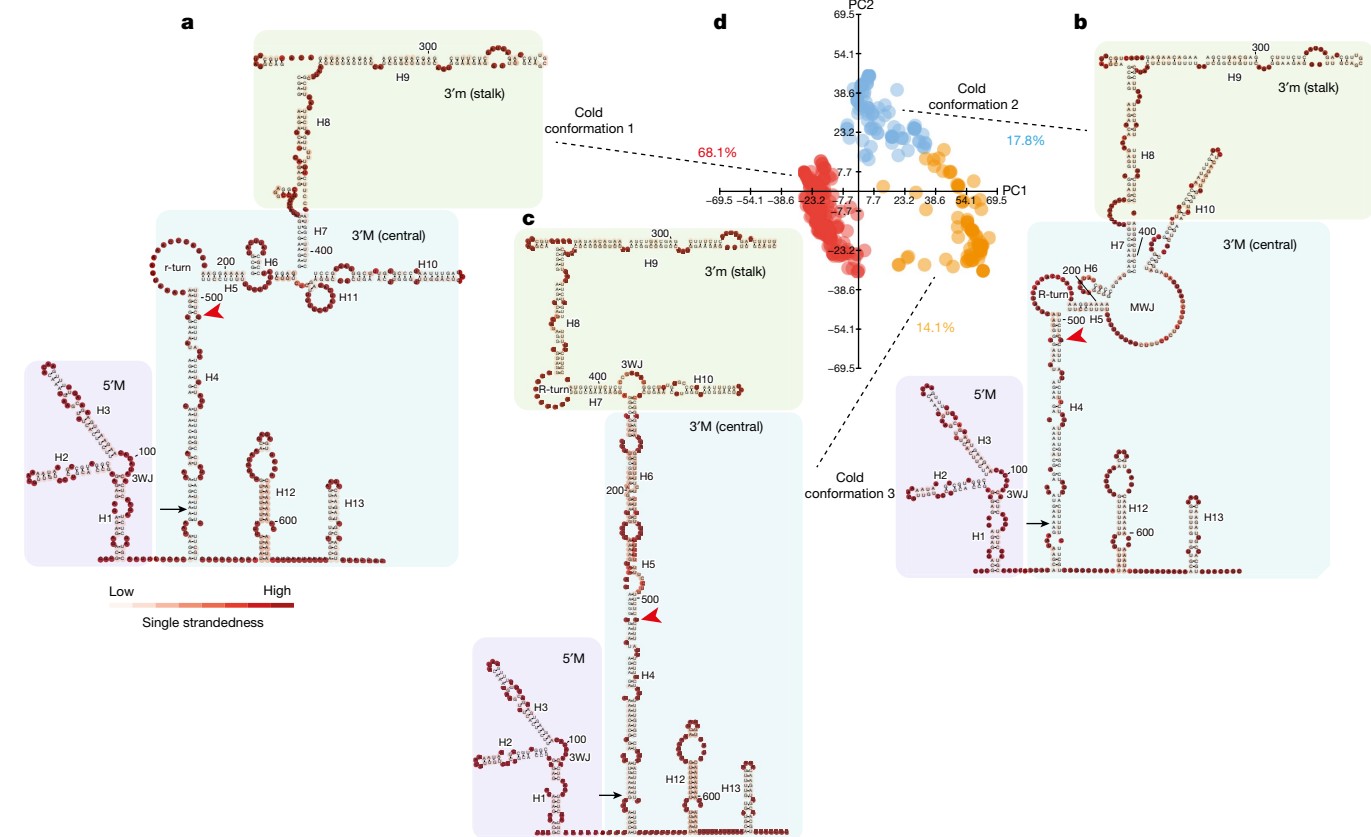

**Fig. 3 | The three major conformations of class II.i in cold-grown plants.**
**a**–**c**, Representative structural models of cold conformation 1 (**a**), cold conformation 2 (**b**) and cold conformation 3 (**c**) from **d**. Models were coloured according to the likelihood of single strandedness. The red arrowheads indicate the site corresponding to the *FLC* TSS. **d**, Visualization of the in vivo structural conformations of class II.i in cold-grown plants. Structures were directly generated from 1,269 individual mutational profiles. Data were visualized using PCAs. Each dot represents a unique single structure derived from each single-molecule mutational profile.

polyadenylated *COOLAIR* found in warm-grown plants lies in the region between H4 and H6, which we term the hyper-variable region; this region is complementary to the sequence of the *FLC* TSS (Extended Data Figs. 4e and 5c).

## *COOLAIR* conformations change in the cold

We then determined *COOLAIR* isoform-specific structures in plants that had been exposed to cold for two weeks. After cold treatment, SHAPE profiles of class I transcripts still showed a low percentage of modification (Fig. 1b) and class II.i was still the most abundant class II isoform (Extended Data Fig. 1a). We identified at least three class II.i conformations (68.1% cold conformation 1; 17.8% cold conformation 2 and 14.1% cold conformation 3 in Fig. 3). Cold conformations 1 and 2 are structurally similar to warm conformations 1 and 2, but their relative proportions are slightly changed (Figs. 2 and 3). Cold-conformation 3 is distinct from warm conformation 3, with the region between H4 and H6 joined into a long stem in cold conformation 3 (Figs. 2 and 3). Taken together, there are two predominant structural conformations of class II.i, the relative proportions of which change in response to cold, with a new conformation emerging in cold-grown plants (cold conformation 3). Comparing the warm-specific (warm conformation 3) and cold-specific (cold conformation 3) structural landscapes of class II.i, the greatest structural difference again occurs in the hyper-variable H4–H6 region complementary to the *FLC* TSS (Extended Data Fig. 4f).

By contrast, the strongly cold-upregulated *COOLAIR* isoform, class II.ii[6], which contains an additional exon compared with class II.i, was found not to adopt major conformations (Extended Data Fig. 6a,b).

An ensemble-averaged structure model for class II.ii revealed four domains (Extended Data Fig. 6a,b), showing the high structural diversity of this isoform as indicated by the high Shannon entropy (Extended Data Fig. 6c,d). This feature might be involved in its functionality associated with the sequestration of FRIGIDA (FRI)[6], the major activator of *FLC* transcription. FRI associates with a range of co-transcriptional regulators related to RNA polymerase II near the *FLC* promoter region in warm conditions and is sequestered, in a class-II.ii-dependent manner, into biomolecular condensates away from the *FLC* promoter after cold exposure[6].

## *COOLAIR* structure–function dissection

Our multiple structural comparisons have identified H4–H6 as a hyper-variable region (Extended Data Figs. 4e,f and 5c). To analyse the potential functional role of this region, we generated transgenic plants where the DNA contained four-nucleotide mutations (mut) designed to increase the bulge in the H4–H6 region by shortening H4 and H5 (Fig. 4a–d and Extended Data Fig. 7a). The structural effect of these four mutations was confirmed by smStructure-seq (Fig. 4d). We then performed a systematic characterization of the *COOLAIR* transcript isoforms in the mut line: the splicing pattern and expression level of *COOLAIR* were not affected (Extended Data Fig. 8a–d). However, the proportion of chromatin-bound class II.i increased in the mut line (Extended Data Fig. 8e), indicating an enhanced interaction between class II *COOLAIR* RNA and *FLC* chromatin. This was confirmed using chromatin isolation by RNA purification (ChIRP), which showed increased chromatin association of the class II *COOLAIR*

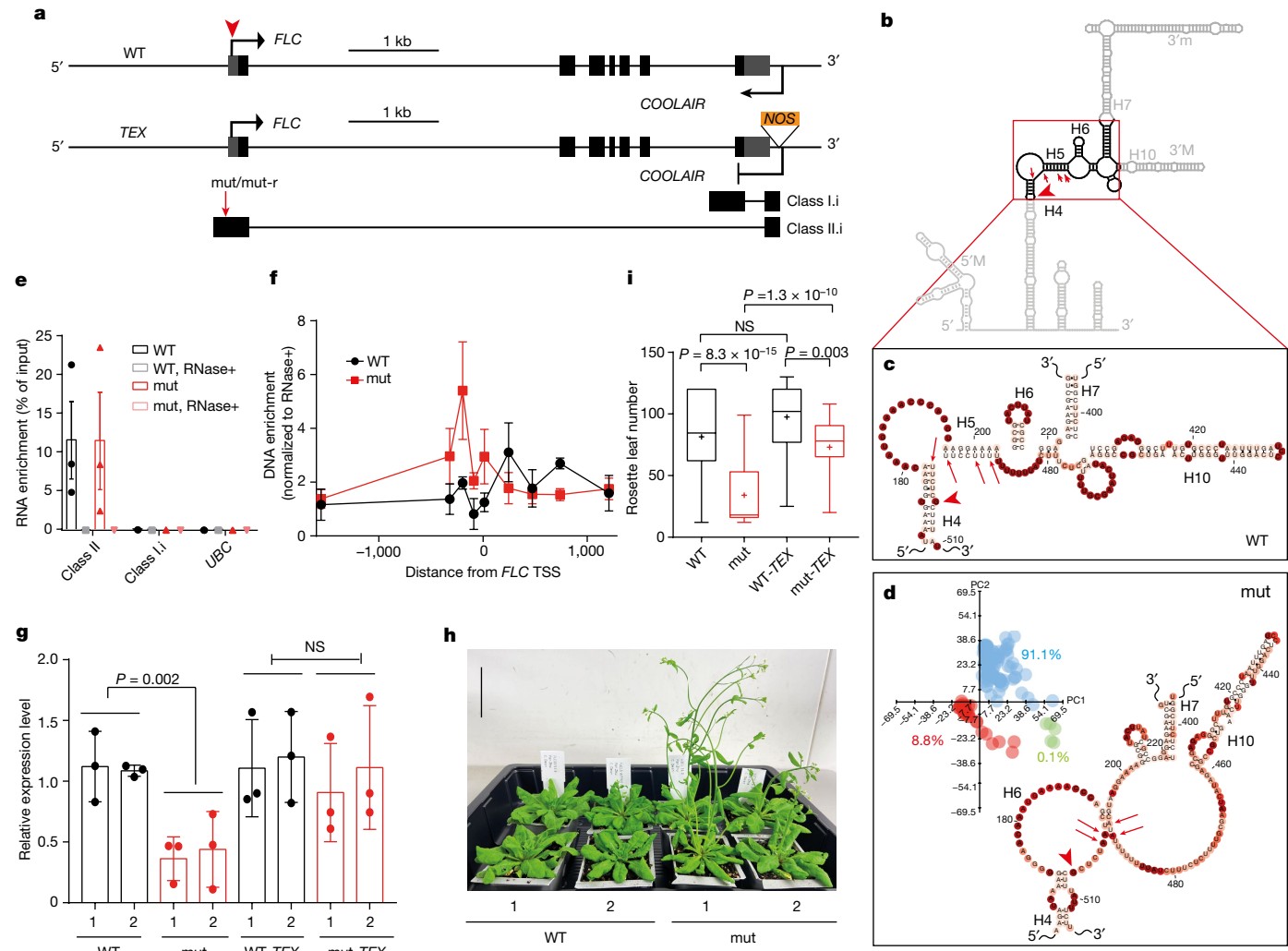

**Fig. 4 | COOLAIR structure-function analysis. a**, Schematic of *FLC* and *COOLAIR* in the wild-type (WT) and *TEX* transgenic lines. Grey boxes, untranslated regions; black boxes, exons. **b**, Schematic of the mutation in the major conformation, warm conformation 1 (Fig. 2a). **c**, The H4–H6 region of class II.i in the wild-type line from Fig. 2a. **d**, The H4–H6 region of class II.i in the mut line. Inset, DaVinci analysis of class II.i in warm-grown mut plants from around 300 individual mutational profiles. The mutation sites are indicated by red arrows in **a**–**d**. The red arrowheads indicate the sites corresponding to the *FLC* TSS in **a**–**d**. **e**, Enrichment of class II RNA by ChIRP–qPCR. Data are mean ± s.e.m.; *n* = 3 biologically independent experiments. Class I and *UBC* RNAs, negative controls. RNase+, RNase A/T1 mix was added during the hybridization. **f**, DNA enrichment at the *FLC* TSS region mediated by class II *COOLAIR* by ChIRP–qPCR. Data are mean ± s.e.m.; *n* = 3 biologically independent experiments. The zero indicates the *FLC* TSS. **g**, The relative expression level of unspliced *FLC* transcript by RT–qPCR in the indicated genotypes in warm conditions. Data are mean ± s.d., *n* = 3 biologically independent experiments. The 1 and 2 indicate independent transgenic lines. **h**, Flowering phenotype of wild-type and mut plants after cold exposure. Scale bar, 50 mm. **i**, Box plots showing the flowering time of the indicated transgenic plants grown in warm conditions measured by rosette leaf numbers. Centre lines show the median, box edges delineate the 25th and 75th percentiles, bars extend to the minimum and maximum values and crosses indicate the mean value. *P* values in **g** and **i** were calculated using a one-way ANOVA. For each genotype, populations of mixed T3 lines are analysed, from left to right, *n* = 36, 35, 36 and 36.

across the *FLC* TSS region in the mut line (Fig. 4e,f). This 5′ ChIRP signal has previously been shown to be sensitive to proteinase K[4]. The mut lines produced lower levels of both unspliced and spliced *FLC* transcript (Fig. 4g and Extended Data Fig. 8f), and were consequently early flowering (Fig. 4h,i). A second mutant (mut-r) in which nucleotides were introduced to decrease the bulge and increase the H4–H6 helix behaved similarly to the wild-type transgene (Extended Data Fig. 7a–c).

Because the introduced mutations were close to the *FLC* TSS, they could potentially influence sense *FLC* transcription activity itself. We therefore introduced the same mutations into a transgene in which antisense *COOLAIR* expression had been disrupted by inserting a *NOS* terminator (*TEX* 2.0)[3] (Fig. 4a). *FLC* transcript levels in mut-*TEX* were similar to those of wild-type *TEX* lines (WT-*TEX*) and higher than those of the mut lines (Fig. 4g and Extended Data Fig. 8f), supporting the

requirement of *COOLAIR* in the flowering time changes induced by the mutations. The necessity of *COOLAIR* to be associated with the chromatin to effect these functional changes was tested by crossing a line carrying the mut transgene with the wild type. Analysis of the F₁ plants enabled us to examine whether *COOLAIR* derived from the mut transgene influenced *FLC* expression of wild-type allele. We found that the *FLC* expression level in F₁ lines was around 50% of that in the wild-type parental line (Extended Data Fig. 8g); therefore, the structural mutations function only on local *FLC* expression. In summary, increasing the bulges around the H4–H6 region promoted a *COOLAIR–FLC* chromatin association, reduced transcriptional output at the *FLC* locus and shortened the time to flower.

Given the complementarity of the H4–H6 region to the *FLC* TSS region, we reasoned that the conformation-dependent *COOLAIR–FLC*

chromatin association might involve the direct binding of *COOLAIR* to *FLC* DNA. Potentially, *COOLAIR* could complement the *FLC* Watson strand to form a DNA–RNA duplex, although we have not found *COOLAIR* to form a significant R-loop at the 5′ end of *FLC*[19]. Alternatively, *COOLAIR* could bind to the double-stranded DNA (dsDNA) to form a DNA–RNA triplex[20,21] (Extended Data Fig. 9a); the sequence content around the H4–H6 region (Fig. 4b,c) is capable of forming triplex structures with the dsDNA at the *FLC* TSS in vitro (Extended Data Fig. 9b). However, because of the proteinase K sensitivity[4] of the ChIRP signal, we favour a model in which *COOLAIR* associates with a protein complex that binds close to the *FLC* TSS. FRI is central to establishing a local chromosomal environment at *FLC*[22], so we tested the involvement of FRI in the functionality of *COOLAIR* conformation by analysing the structurally mutated transgene (mut) in both active *FRI* and null *fri* genotypes (Extended Data Fig. 8h). Structural mutations influence *FLC* expression in only the *FRI* genotype (Extended Data Fig. 8h). Therefore, in addition to the physical association of FRI with *COOLAIR* class II.ii in cold conditions, the structurally variable region of *COOLAIR* class II.i genetically interacts with FRI to regulate *FLC* expression in warm conditions. How the individual *COOLAIR* structural conformations of the different isoforms affect *FLC* transcription will be an exciting future area of investigation.

In summary, development of the single-molecule-based RNA structure profiling methodology has allowed us to directly determine the in vivo RNA structure of the antisense transcripts of *COOLAIR*. This methodology has enabled the structural conformations of each alternatively processed *COOLAIR* isoform to be described. In response to cold conditions, the proportion of *COOLAIR* adopting a certain conformation changes and new conformations emerge. Across the whole structural landscape of *COOLAIR*, we identified a structural element that showed the greatest conformational variation, which was complementary to the *FLC* TSS. We validated a functional role for this structural element in regulating *COOLAIR*–*FLC* chromatin association, *FLC* expression and flowering time, suggesting a functional role for RNA conformational changes in the environmental response of plants[5,6,23–25]. Our study provides insights into how lncRNA transcript isoforms can adopt different RNA structural conformations, and how these can functionally influence the association with chromatin and control transcription.

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

## Methods

### Statistics

No statistical methods were used to predetermine the sample size. The experiments were not randomized, and investigators were not blinded to allocation during experiments and outcome assessment. Sampling in all cases was performed by collecting materials independently from separate plants.

### Plant materials and growth conditions

The genotypes Col *FRI*[SF2] (Col *FRI*) and Var2–6 near-isogenic line have been described previously[3,7]. *FLC*[WT], *FLC*[WT]*-TEX*, *FLC*[mut], *FLC*[mut-r], *FLC*[mut]*-TEX* and *FLC*[mut-r]*-TEX* were transgenic lines carrying an approximately 12 kb wild-type or mutated *FLC* genomic fragment. *FLC*[mut] was generated by introducing four-nucleotide mutations using site-directed mutagenesis. *FLC*[WT]*-TEX* and *FLC*[mut]*-TEX* were generated by inserting a *NOS* terminator fragment in the first exon of *COOLAIR* in the wild-type or mutated *FLC* genomic fragment, respectively[3]. *FLC*[mut-r] was generated by inserting a fragment (GAAATAAAGCGAGAACAAAT-GAAAACCCAGGT) complementary to the big bulge in the H4–H6 region using site-directed mutagenesis. Primers used for the construction are listed in Supplementary Table 1. The fragments were then cloned into SLJ77515 (ref. [26]) and transformed into the *Arabidopsis flc-2 FRI* genotype[3] with a floral-dipping method. Transgenic lines with a single insertion that segregated 3:1 for Basta resistance were identified in the $T_2$ generation to generate homozygous $T_3$ lines. $T_3$ homozygous lines with *FLC*[mut] in *flc-2 FRI* background were crossed with Col *FRI* (WT) for $F_1$ generation (Extended Data Fig. 8g) or with the *flc-2 fri* background for *FLC*[mut] *fri* (Extended Data Fig. 8h).

Seeds were surface-sterilized and sown on half-strength Murashige and Skoog medium. The plates were kept at 4 °C for 2–3 days. For warm-grown plants, seedlings were grown in warm conditions (16 h light, 8 h darkness with constant 20 °C) for 10 days. For the cold treatment, the plants were subjected to a two-week treatment at 5 °C (8 h light and 16 h dark conditions) after a 10-day pre-growth period in warm conditions.

### (+)SHAPE and (−)SHAPE smStructure-seq library construction

We used the SHAPE reagent, NAI, to do the in vivo RNA secondary structure chemical probing. NAI was prepared as reported previously[13]. In brief, *A. thaliana* seedlings were completely covered in 20 ml 1× SHAPE reaction buffer (100 mM KCl, 40 mM HEPES (pH 7.5) and 0.5 mM MgCl₂) in a 50-ml Falcon tube. NAI was added to a final concentration of 1 M and the tube swirled on a shaker (1,000 rpm). This high NAI concentration allows NAI to penetrate plant cells and modify the RNA in vivo. After quenching the reaction with freshly prepared dithiothreitol (DTT), the seedlings were washed with deionized water and immediately frozen with liquid nitrogen and ground into powder. Total RNA was extracted using the hot phenol method[4], followed by DNase I treatment in accordance with the manufacturer's protocol. The control group was prepared using DMSO (labelled as (−)SHAPE), following the same procedure as described above. Then, 2 μg (+)SHAPE or (−)SHAPE RNA samples was added to a 19-μl buffer system containing 2 μl 0.5 μM RNA–DNA hybrid adaptors (5′-rArGrArUrCrGrGrArArGrArGrCr ArCrArCrGrUrCrUrGrArArCrUrCrCrArGrUrCrArC/3SpC3/ and 5′-GTGACTGGAGTTCAGACGTGTGCTCTTCCGATCTN (N = equimolar A, T, G, C)), 4 μl 5× reaction buffer (2.25 M NaCl, 25 mM MgCl₂, 100 mM Tris-HCl, pH 7.5), 2 μl 10× DTT (50 mM; made fresh or from frozen stock) and 1 μl TGIRT-III enzyme (10 μM; InGex). The reaction system was pre-incubated at room temperature for 30 min, then 1 μl of 25 mM dNTPs (an equimolar mixture of dATP, dCTP, dGTP and dTTP; at 25 mM each; RNA-grade) was added. The whole reaction system in the tube was incubated at 60 °C for 120 min. To remove the TGIRT-III enzyme from the template, 1 μl of 5 M NaOH was added and the sample incubated at 95 °C for 3 min. The sample was cooled down to room temperature

and neutralized with 1 μl of 5 M HCl before the clean-up of the cDNAs with a MinElute Reaction Cleanup Kit (QIAGEN, 28204). To capture class I and class II *COOLAIR* isoforms along with 18S rRNA, PCR reactions with 10 cycles were done with specific primers (Supplementary Table 1) using KOD Xtreme Hot Start DNA Polymerase (Novagen). The amplified DNA fragments from the eight replicates of the PCR reactions were merged to obtain sufficient DNA. The resulting DNA samples were size-selected using the Solid Phase Reversible Immobilization size-selection system (BECKMAN COULTER). Two independent biological replicates were generated for both (+)SHAPE and (−)SHAPE smStructure-seq libraries. The purified DNA samples were subjected to PacBio library construction by BGI using a PacBio Sequel 3.0.

### smStructure-seq data analysis of *COOLAIR* isoforms

The raw reads from (+)SHAPE and (−)SHAPE libraries were converted into HiFi reads (circular consensus sequences) using 'ccs' (https://github.com/PacificBiosciences/ccs) with parameters '--minPasses=3' in order to achieve around 99.8% predicted accuracy (Q30)[14]. The HiFi reads were demultiplexed using the demultiplex barcoding algorithm Lima v.1.11.0 (https://github.com/pacificbiosciences/barcoding). The derived HiFi reads were mapped to both *COOLAIR* references and 18S rRNA (Supplementary Table 1) using BLASR (v.5.3.3)[27] with parameters '--minMatch 10 -m 5 --hitPolicy leftmost'. Each read was converted into a 'bit vector'. In brief, each bit vector corresponds to a single read and consists of series of zeroes (representing matches) and ones (mutations representing mismatches and unambiguously aligned deletions)[11]. To generate the overall SHAPE reactivity profiles, the mutation rate (MR) at a given nucleotide is simply the total number of ones divided by the total number of zeroes and ones at that location. Raw SHAPE reactivities of class II *COOLAIR* were then generated for each nucleotide using the following equation:

$$R = \frac{MR_{(+)SHAPE} - MR_{(-)SHAPE}}{1 - MR_{(-)SHAPE}}$$

where (+)SHAPE corresponds to a NAI-treated sample and (−)SHAPE refers to a DMSO-treated sample. The true-negative rate, $1 - MR_{(-)SHAPE}$, represents the specificity at a specific location. The raw SHAPE reactivity ($R$) mathematically estimates the positive likelihood ratio of SHAPE modification. The raw SHAPE reactivity was normalized to a standard scale that spanned from 0 (no reactivity) to around 1 (high SHAPE reactivity)[28] for showing the mutational profiles.

### Structural analysis of class II *COOLAIR* isoforms by DaVinci

The whole pipeline of DaVinci is illustrated in Extended Data Fig. 2a. The bitvectors generated from previous step were transformed into constraint information ('1' representing single-stranded nucleotides) for each sequencing read of class II *COOLAIR* isoforms. The single-stranded constraints were incorporated into the SCFG engine of the DaVinci pipeline. The SCFG engine, including a set of transformation rules for SCFG and a probability distribution of the transformation rules for each non-terminal symbol, was provided by CONTRAfold[29] with an extended function utility in CentroidFold[30] (--engine CONTRA-fold --sampling). The generated RNA structures with constraints derived from individual bitvectors were collected. Because the different structures can have the same mutational profile during probing, we used the sampling function with constraint of a bitvector to capture multiple structures of class II.ii *COOLAIR* isoforms. All of the collected RNA structures were transformed into dot-bracket strings followed by transformation into RNA structure elements using rnaConvert in the Forgi package[31]. The digitalized RNA secondary structure elements were extracted to create a numeric matrix and subjected to dimensionality reduction, such as PCA or multidimensional scaling. The dimensionality reduction results were clustered using $k$-means clustering with the $k$-means function from the scikit-learn Python package[32]. The

value of $k$ was set as determined visually. The representative structure for each cluster was identified by calculating the most common RNA structure type at each position (that is, the maximum expected accuracy) and was determined by the RNA structure that is at the centre of the cluster and most similar to the most common RNA structure. The base-pair probability was calculated by counting the frequency of all present base pairs in the conformation space. The positional base-pair probability was derived by $P_i = \sum_j^J P_{ij}$, where $P_{ij}$ is the probability of base $i$ of being base-paired with base $j$, over all its potential $J$ pairing partners. The likelihood of single strandedness was calculated by the expression of $1 - P_i$. In addition, the Shannon entropy was calculated as $E_i = \sum_j^J - P_{ij} \log_{10}(P_{ij})$.

### Structural analysis of HIV-1 RRE, RRE61, cspA and TenA
Probing data for HIV-1 RRE[11] were obtained from RRE-invitroDMS_NL43rna.bam (https://codeocean.com/capsule/6175523/tree/v1). Probing data for the cspA 5′ untranslated region[33] at 37 °C and 10 °C were obtained from Sequence Read Archive (accessions numbers SRR6123773 and SRR6123774). We performed the RNA structure probing experiments of in vitro folded HIV-1 RRE61 RNAs (3 pmol) containing the stem loops III, IV and V[18] as described previously[11]. The TenA RNAs (3 pmol) were subjected to NAI chemical treatment[13,34] in the presence or absence of 1 μM thiamine pyrophosphate (TPP). The NAI-modified RNA samples (TPP-treated and non TPP-treated RNAs) were mixed with a ratio of 20:80 (vol/vol) or 50:50 (vol/vol) for the library construction. All of the sequencing data were mapped to the respective references as described above. The subsequent bitvectors were generated and subjected to the DaVinci analysis described above, including the creation of the numeric matrix for the digitalized RNA structure elements, dimensionality reduction, $k$-mean determination and representative structure construction. In silico structural ensemble analysis of RRE wild-type and mutant RRE61 were performed by Boltzmann sampling (10,000 times) using RNAfold[35]. The subsequent analysis for the in silico structure ensemble is the same as for the DaVinci analysis but includes only the steps of creating the numeric matrix for the digitalized RNA structure elements, dimensionality reduction, $k$-mean determination and representative structure construction.

### Total RNA extraction and RT–qPCR for gene expression analysis
Total RNA was extracted as previously described[36]. Genomic DNA was digested with TURBO DNA-free (Ambion Turbo DNase kit, AM1907) according to the manufacturer's guidelines before reverse transcription was performed. Reverse transcription was performed with the SuperScript III Reverse Transcriptase (ThermoFisher, 18080093) following the manufacturer's protocol using gene-specific primers. The standard reference gene *UBC* (At5g25760) for gene expression was used for normalization. All primers are listed in Supplementary Table 1.

### Chromatin-bound RNA measurement assay
Chromatin-bound RNAs were extracted as previously outlined[37]. In brief, 2 g of warm-grown or cold-grown seedlings were ground into fine powder using mortar in liquid nitrogen. Then, 1% of the materials (about 200 mg fine powder) was used for total RNA extraction as described above. The nuclei from the remaining material were prepared with Honda buffer in the presence of 50 ng μl$^{-1}$ tRNA, 20 U ml$^{-1}$ RNase inhibitor (SUPERase-In; Life Technologies), and 1× cOmplete protease inhibitor (Roche). The nuclei pellet was resuspended in an equal volume of resuspension buffer (50% (vol/vol) glycerol, 0.5 mM EDTA, 1 mM DTT, 100 mM NaCl and 25 mM Tris-HCl pH 7.5) and washed twice with urea wash buffer (300 mM NaCl, 1 M urea, 0.5 mM EDTA, 1 mM DTT and 1% Tween-20 and 25 mM Tris-HCl pH 7.5). Two volumes of wash buffer were added to the resuspended nuclei and vortexed for 1 s. The chromatin was spun down and protein was removed using phenol–chloroform. RNAs from the supernatant were precipitated with isopropanol, dissolved and DNase-treated.

The chromatin-bound RNAs were reverse-transcribed with the SuperScript III Reverse Transcriptase (ThermoFisher, 18080093) following the manufacturer's protocol. A mixture of gene-specific primers (Supplementary Table 1) and *EF1alpha* (At5g60390.2)[37,38], to estimate how many RNAs were bound to genome DNA (expressed as (chromatin-bound RNA)/*EF1alpha*), were included in the reverse-transcription reaction. The total RNAs were also reverse transcribed with the SuperScript III Reverse Transcriptase (ThermoFisher, 18080093) following the manufacturer's protocol. A mixture of gene-specific primers (Supplementary Table 1) and *PP2A* (At1g13320) as a control were added to the reverse-transcription reaction, which estimates the total expression level of class II (expressed as (total RNA)/*PP2A*). The chromatin-binding ratio was calculated using the equation:

$$\text{Chromatin–binding ratio} = \frac{(\text{Chromatin–bound RNA})/EF1alpha}{(\text{Total RNA})/PP2A}.$$

### ChIRP–qPCR assay
ChIRP was performed as previously outlined, with some modifications[4,39,40]. Antisense DNA probes were designed against the distal exon sequence of *COOLAIR* class II and biotinylated at the 3′ end; probes are listed in Supplementary Table 1. Then, 3 g of warm-grown seedlings were crosslinked in 3% (vol/vol) formaldehyde at room temperature in a vacuum. Crosslinking was then quenched with 0.125 M glycine for 5 min. Crosslinked plants were ground into a fine powder and lysed in 50 ml of cell lysis buffer (20 mM Tris-HCl pH 7.5, 250 mM sucrose, 25% glycerol, 20 mM KCl, 2.5 mM MgCl$_2$, 0.1% NP-40 and 5 mM DTT). The lysate was filtered through two layers of Miracloth (Merck, D00172956) and pelleted by centrifugation. The pellets were washed twice with 10 ml of nuclear wash buffer (20 mM Tris-HCl pH 7.5, 2.5 mM MgCl$_2$, 25% glycerol, 0.3% Triton X-100 and 5 mM DTT). The nuclear pellet was then resuspended in nuclear lysis buffer (50 mM Tris-HCl pH 7.5, 10 mM EDTA, 1% SDS, 0.1 mM PMSF and 1 mM DTT) and sonicated using a Bioruptor ultrasonicator (Diagenode). All of the buffers were supplemented with 0.1 U μl$^{-1}$ RNaseOUT (Life Technologies), 1 mM PMSF and Roche cOmplete tablets to keep the integrity of any RNA–protein and protein–protein complexes. The following steps were performed as previously described[40]. For each reaction, 30 μl pre-blocked Streptavidin C1 magnetic beads (Thermo Fisher Scientific, 65001) were used. Then, 20 μl of RNase A/T1 Mix (Thermo Fisher Scientific, EN0551) instead of RNaseOUT was added into the RNase+ reactions (Fig. 4e), just before the hybridization (at 37 °C for 4 h) started; these samples were used as the control for background noise. RNA was eluted and reverse transcribed using SuperScript IV Reverse Transcriptase (ThermoFisher, 18090050) with gene-specific primers. *COOLAIR* enrichment and DNA eluted was analysed by RT–qPCR. All primers used for reverse transcription and RT–qPCR are listed in Supplementary Table 1.

### Electrophoretic mobility shift assays
Electrophoretic mobility shift assays (EMSAs) were performed as described previously[21] using oligonucleotides end-labelled with Cy5 (DNA) or FAM (RNA). Oligonucleotide sequences are shown in Supplementary Table 1. EMSAs were done using home-made 15% polyacrylamide gels with 40 mM Tris-acetate (pH 7.4) and 10 mM MgCl$_2$ at 15 volt cm$^{-1}$. Gel images were taken with a Typhoon FLA 9500 fluorescence reader (GE Healthcare Life Sciences). Sequences for the positive control rDNA enhancer En3-PAPAS were obtained from a previous study[21].

### Reporting summary
Further information on research design is available in the Nature Research Reporting Summary linked to this article.

## Data availability

Sequencing data have been deposited in the Sequence Read Archive (SRA) under BioProject ID number PRJNA749291. A full list of DNA oligomers, PCR primers and *COOLAIR* reference sequences is available in Supplementary Table 1. The raw data of RNA-expression level, RT–qPCR and ChIRP–qPCR that support the findings of this study are available as Source Data. Uncropped images of EMSA and RT–qPCR are available in Supplementary Fig. 1. Accession numbers (from The Arabidopsis Information Resource (TAIR; https://www.arabidopsis.org/)) for the genes analysed in this study are *FLC* (*At5g10140*) and *COOLAIR* (*At5g01675*). Standard reference genes *EF1alpha* (*At5g60390*), *PP2A* (*At1g13320*) and *UBC* (*At5g25760*) for gene expression were used for normalization. Source data are provided with this paper.

## Code availability

Code is publicly available at GitHub (https://github.com/DingLab-RNAstructure/smStructure-seq).

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

**Acknowledgements** This work was funded by the European Research Council (grant 680324; to Y.D.), a Wellcome Senior Investigator (grant 210654; to C.D.), a Royal Society Professorship (RP\R1\180002; to C.D.), by the Biotechnology and Biological Sciences Research Council (BB/L025000/1; to Y.D.); and by Institute Strategic Programmes GRO (BB/J004588/1) and GEN (BB/P013511/1) to Y.D. and C.D.

**Author contributions** M.Y., C.D. and Y.D. conceptualized the study. M.Y., P.Z., C.D. and Y.D. wrote the paper. Q.L., P.Z. and R.B. performed the SHAPE probing and RNA extraction. R.B. generated *COOLAIR* structural mutation constructs and transgenic plants. P.Z. performed the phenotypic analysis, gene-expression and genetic studies as well as the ChIRP assay of the structural mutants. M.Y. and Y.Z. constructed the RNA structure libraries. P.M. performed triplex EMSA experiments. M.Y. and J.C. analysed the sequencing data. C.D. and Y.D. acquired funding. C.D. and Y.D. conducted the project administration. C.D. and Y.D. supervised the study.

**Competing interests** A patent application (LU501541) naming Y.D., M.Y., J.C. and Y.Z. has been filed by the John Innes Centre for the technology described in this paper.

**Additional information**
**Correspondence and requests for materials** should be addressed to Caroline Dean or Yiliang Ding.

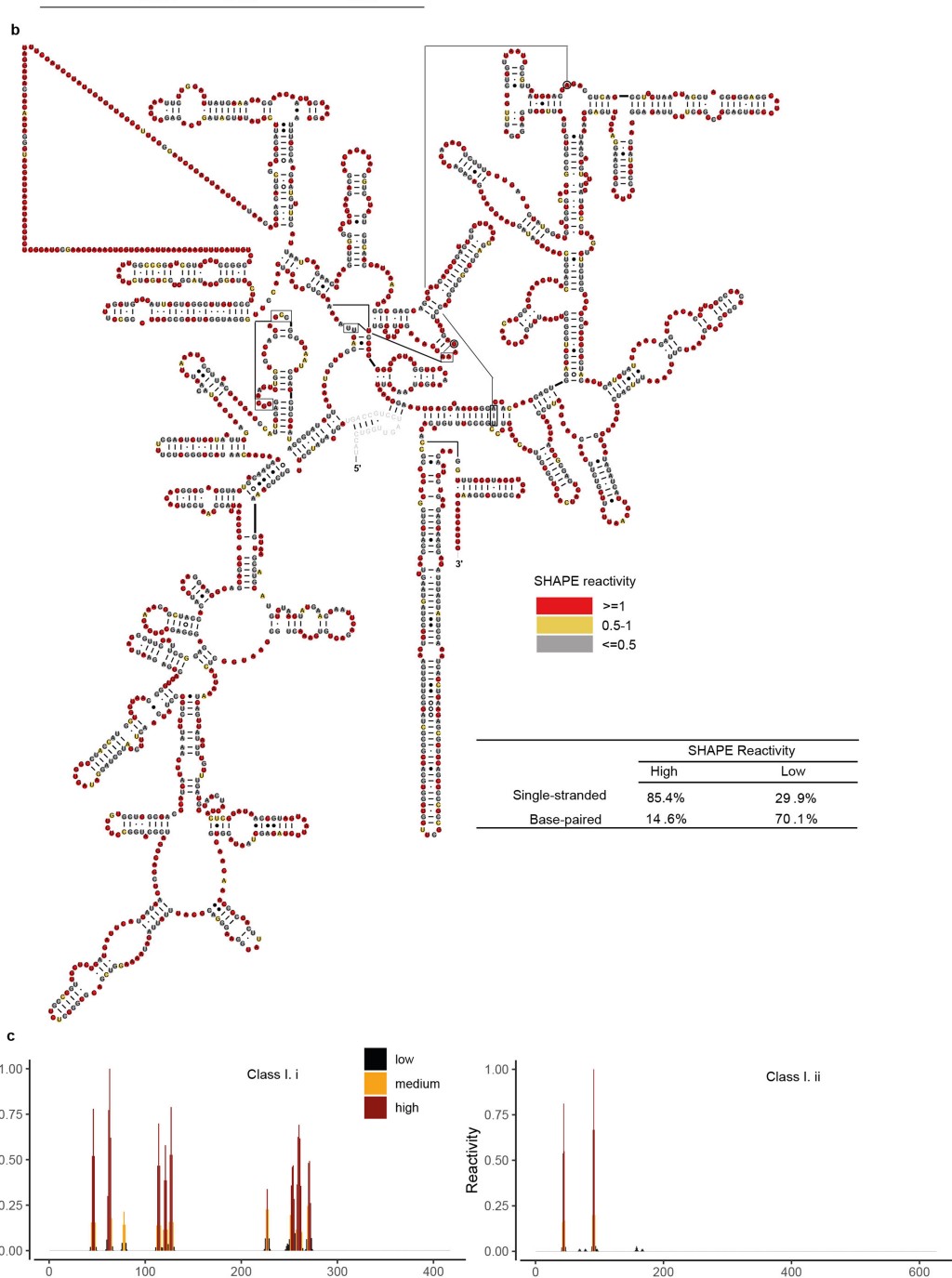

**a**

|  |  | Warm | Cold |
|---|---|---|---|
| **Class I** | Class I.i | 81.2% | 99.7% |
|  | Class I.ii | 18.8% | 0.3% |
| **Class II** | Class II.i | 96.4% | 97.9% |
|  | Class II.ii | 3.2% | 2.0% |
|  | Others | 0.4% | 0.1% |

**b**

SHAPE reactivity

| | |
|---|---|
| ■ (red) | >=1 |
| ■ (yellow) | 0.5-1 |
| ■ (grey) | <=0.5 |

|  | SHAPE Reactivity | |
|---|---|---|
|  | High | Low |
| Single-stranded | 85.4% | 29.9% |
| Base-paired | 14.6% | 70.1% |

**c**

Class I. i

Class I. ii

low
medium
high

**Extended Data Fig. 1** | See next page for caption.

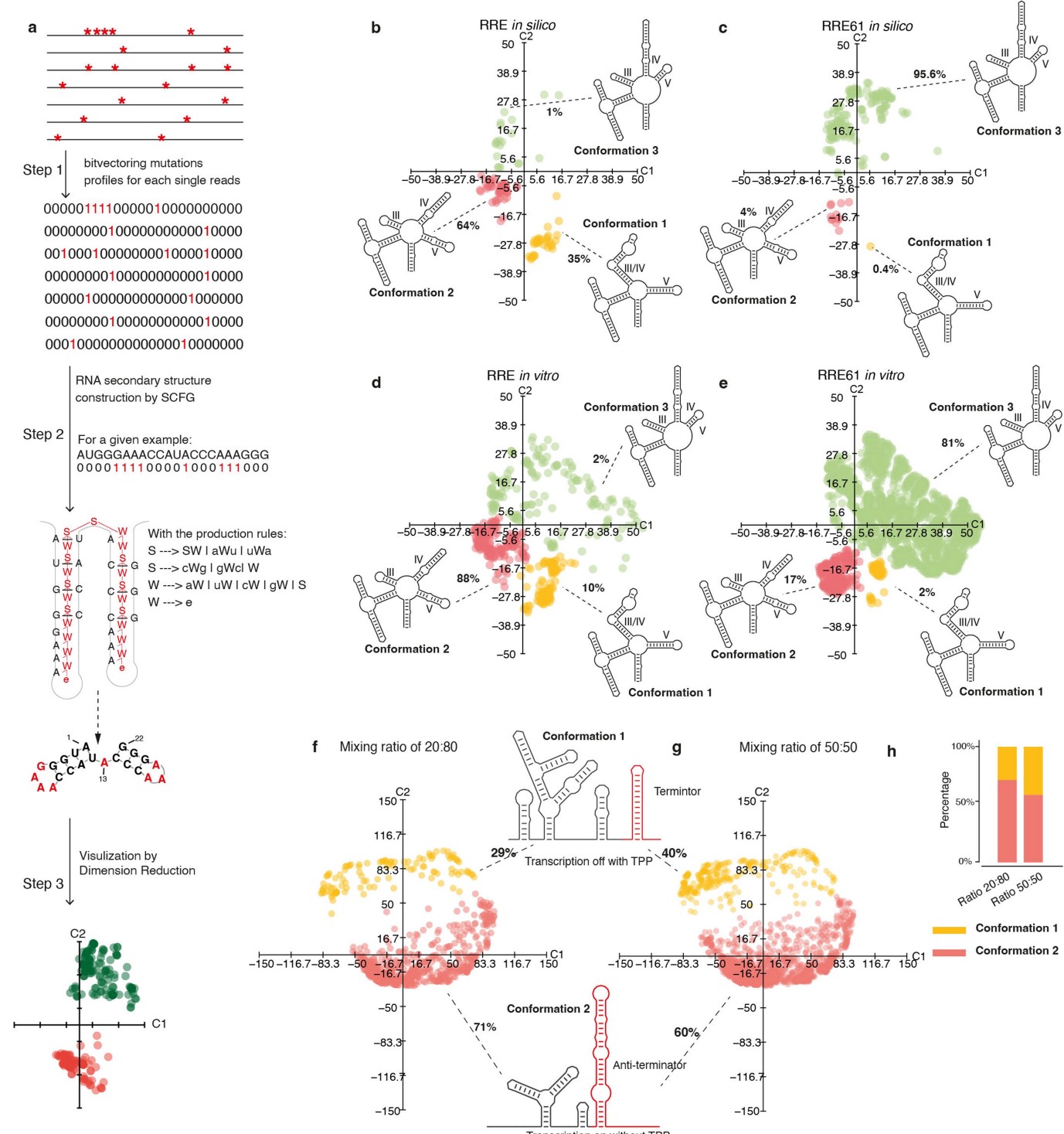

**Extended Data Fig. 1 | smStructure-seq can accurately probe the full-length RNA secondary structure *in vivo*. a**, Percentage of HiFi reads of main *COOLAIR* isoforms from our smStructure-seq libraries was calculated. -27.39 billion total bases (40,123,867 raw reads) of *COOLAIRs* were obtained for both warm and two weeks cold (2W) conditions. **b**, In parallel, we also obtained around 8.58 billion total bases (1,317,882 raw reads) of 18S rRNA which serves as the internal control for our smStructure-seq libraries. The complete 18S rRNA (length 1,808 nt) phylogenetic structure is colour-coded according to the SHAPE reactivity generated from our smStructure-seq (SHAPE reactivity > = 1 marked in red; SHAPE reactivity 0.5–1 marked in yellow; SHAPE reactivity < = 0.5 marked in grey; the unresolved region at the 5'end was labelled as grey colour).

The table quantifies the correspondence between the 18S rRNA phylogenetic structure and the SHAPE reactivities. In the full-length 18S rRNA, 85.4% of nucleotides that show high *in vivo* SHAPE reactivity in our data set correspond to single-stranded regions in the phylogenetic structure (true positive), whereas 70.1% of the nucleotides that show low *in vivo* SHAPE reactivity correspond to base-paired regions in the phylogenetic structure (true negative). Both true positive (85.4%) and true negative (70.1%) signals are much higher than our previous illumina-based short reads method[41]. **c**, Normalized SHAPE reactivities derived from the mutation rate profiles were plotted for Class I *COOLAIR* transcript isoforms under warm-grown condition. The normalized SHAPE reactivity was colour-coded and shown on the Y axis.

**Extended Data Fig. 2 | DaVinci conformation analysis pipeline and its validation. a**, DaVinci conformation analysis pipeline. Each line is referring to one sequencing read. The red stars denote the mutations including mismatch and deletions. In step 1, the sequencing reads are bit-vectorized following the rules: "0" if a base is wild type and "1" if the base is mutated. In step 2, SCFG[42] was applied to derive the RNA structures that can best-represent each mutation profile. For example, given sequence "AUGGGAACCAUACCCAAAGGG" with a bitvector of "0001110000100011000", the production rule (showed in the step 2) can derive the RNA structures as showed in step 2, independent of thermodynamic parameters. "|" in the rules represents the logic of "or" between production rules. The red "1"s or letters indicate the mutation information or single-stranded nucleotides. In step 3, the collected RNA structures derived from each individual mutation profile were transformed into numeric matrix of RNA structure element and subjected to dimensionality reduction. Then, the representative RNA structures for each conformational cluster were determined. Detailed description was in the Methods section. **b**–**e**, The *in silico* (**b**, **c**) and *in vitro* (**d**, **e**) RNA conformational landscape of HIV-1 Rev response element (RRE) region in wild-type sequence (RRE) or mutant RRE61. **f**, Davinci-determined RNA conformation landscape for TenA RNAs folded with and without TPP ligands. The folded RNAs were probed *in vitro* and pooled with the ratio of 20 (TPP-treated RNAs):80 (non TPP-treated RNAs). **g**, Similar to (**f**) but with the pooling ratio of 50 (TPP-treated RNAs):50 (non TPP-treated RNAs). The detailed discussions of (**b**–**g**) were in Supplementary Discussion. **h**, Proportions for each cluster detected by DaVinci. The ratios are derived from (**f**, **g**).

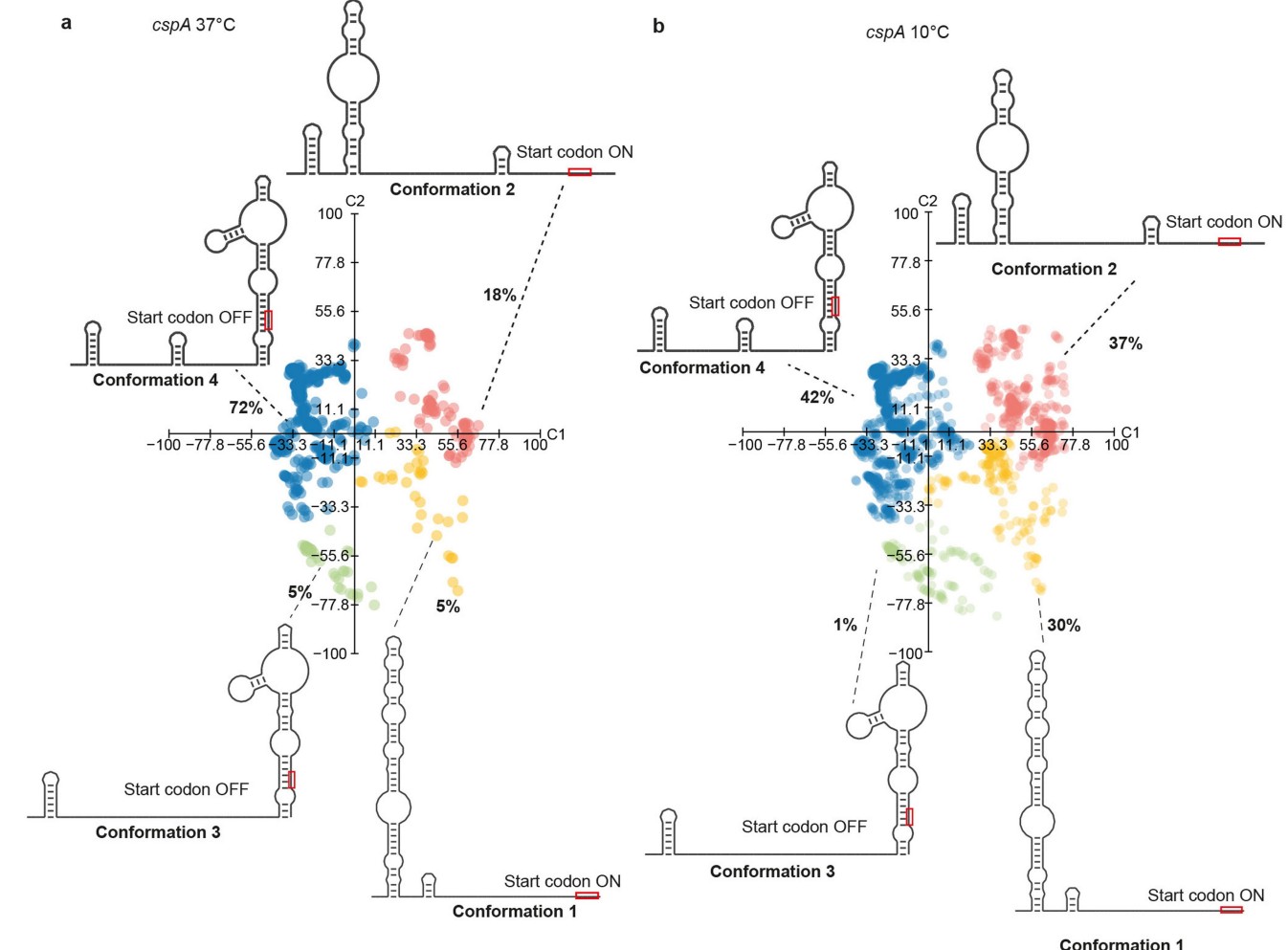

**Extended Data Fig. 3 | The validation of DaVinci conformation analysis.** Davinci-determined RNA structure conformation space for cspA RNA at 37 °C[33] (**a**) or 10 °C[33] (**b**). The red rectangle is the start codon. The detailed discussions were in Supplementary Discussion.

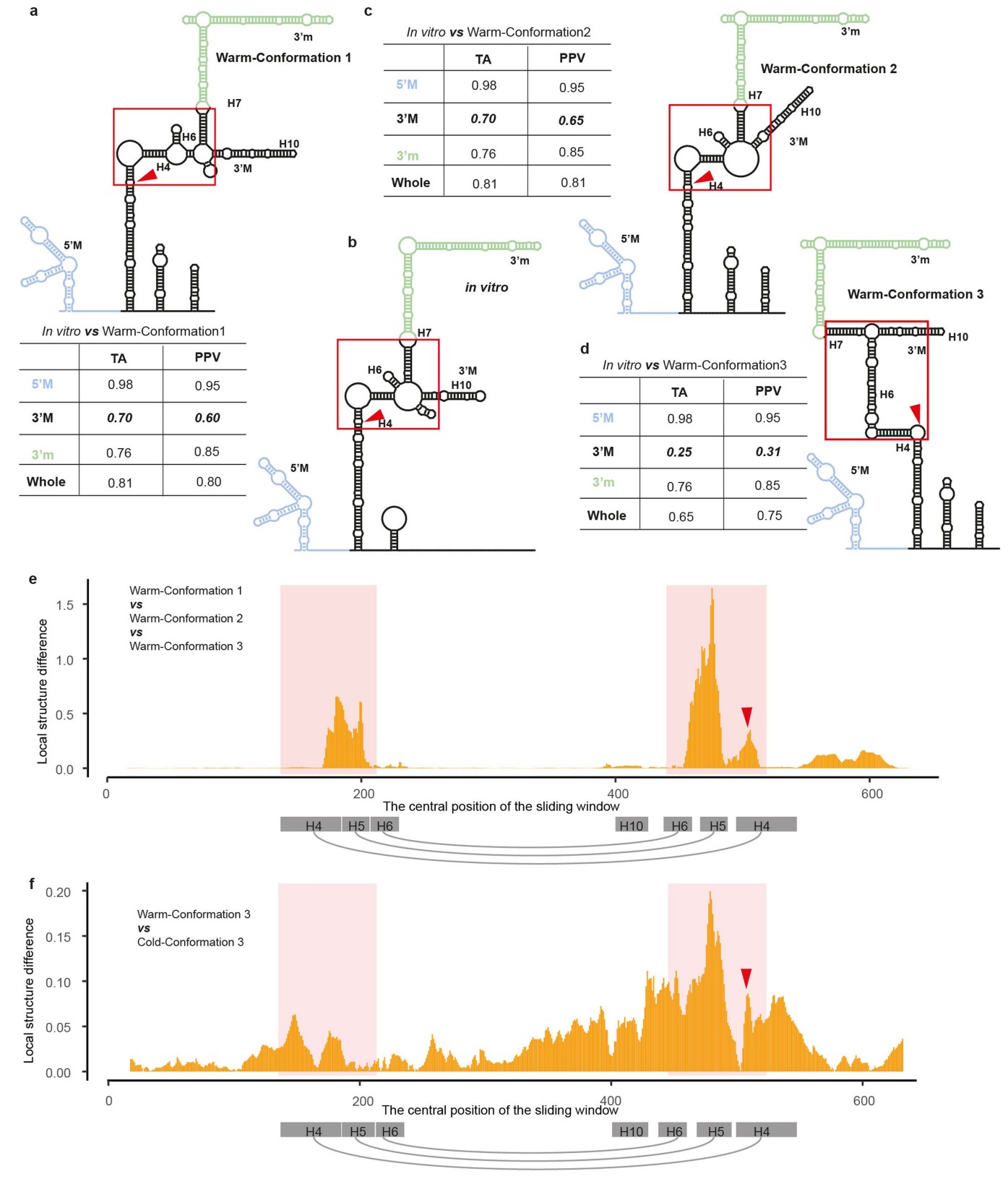

**Extended Data Fig. 4** | See next page for caption.

**Extended Data Fig. 4 | Identification of hyper-variable RNA structural region. a**, RNA structure model of warm conformation 1 in (Fig. 2a). The red triangle indicates the site corresponding to *FLC* TSS. The table is the similarity comparison between each domain as well as the whole structure of warm conformation 1 and the previously reported *in vitro*[5] RNA structure (**b**). The topological similarity was based on the tree alignment (TA)[43] and calculated by RNAforester[35]. The base-pairing similarity was calculated using positive predictive value (PPV)[44]. The red square is showing the dramatic structure change between *in vitro* and *in vivo* RNA structure. **b**, RNA structure model of previously reported *in vitro*[5] Class II.i RNA structure. **c**, **d**, RNA structure models of warm conformations 2 and 3 in (Fig. 2a). The similarity comparison of TA and PPV were listed in the table respectively. The central 3' M domain showed the lowest topological and base-pairing similarities (bold and oblique) between *in vitro* Class II.i and warm conformations 1, 2 and 3. **e**, The local structural difference was measured by the -log$_{10}$($P$ value) in two-way ANOVA test of single-strandedness among these three warm conformations in a sliding window of 30 nt. The red shadow region indicated the greatest difference, i.e, hyper-variable region, among warm conformations 1, 2 and 3. The red shadow regions are corresponding to the red squares in (**a**–**d**). The sequences in H4, H5 and H6 helix were draw in grey rectangle with grey arch indicating the helix formation. **f**, The local structural difference was measured by the -log$_{10}$($P$ value) in the t-test of single-strandedness between the warm specific conformation (warm conformation 3) and cold specific conformation (Cold-Conformation 3). The red shadow regions are corresponding to the red squares in (**a**–**d**).

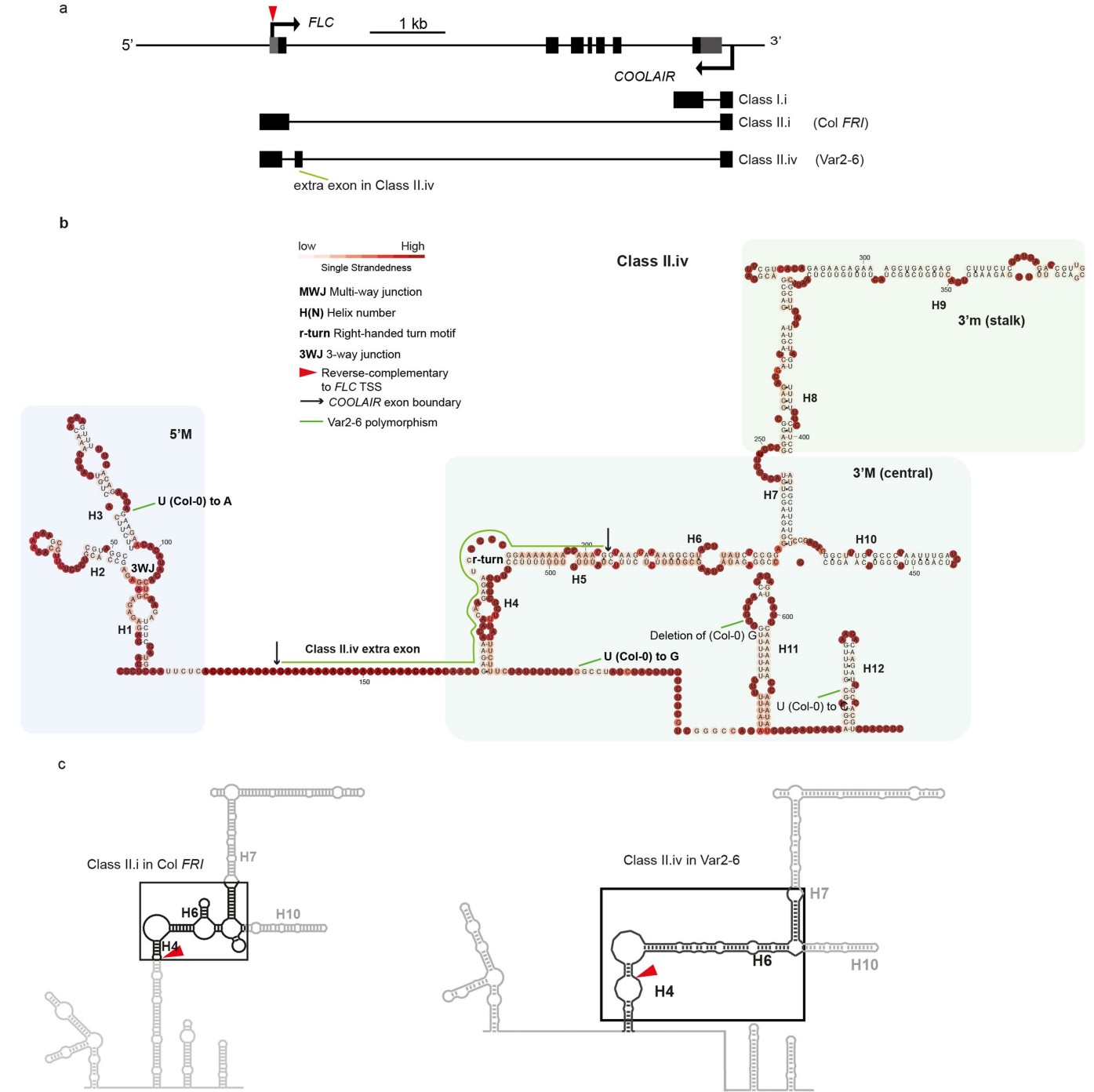

**Extended Data Fig. 5 | Natural variation increases the abundance of a structurally distinct *COOLAIR* isoform. a**, Schematic illustration of *FLC* and *COOLAIR* gene structure. Untranslated regions are shown in grey boxes and exons in black boxes. The extra exon of class II.iv in Var2–6 (ref. [7]) is indicated by a green line. kb, kilobase. The red triangle indicates the site corresponding to the *FLC* TSS. **b**, In vivo structure of class II.iv in the Var2–6 line. **c**, The RNA structural model of warm conformation 1 is from Fig. 4b and Class II.iv from **b**. The hyper-variable regions are shown in the black square.

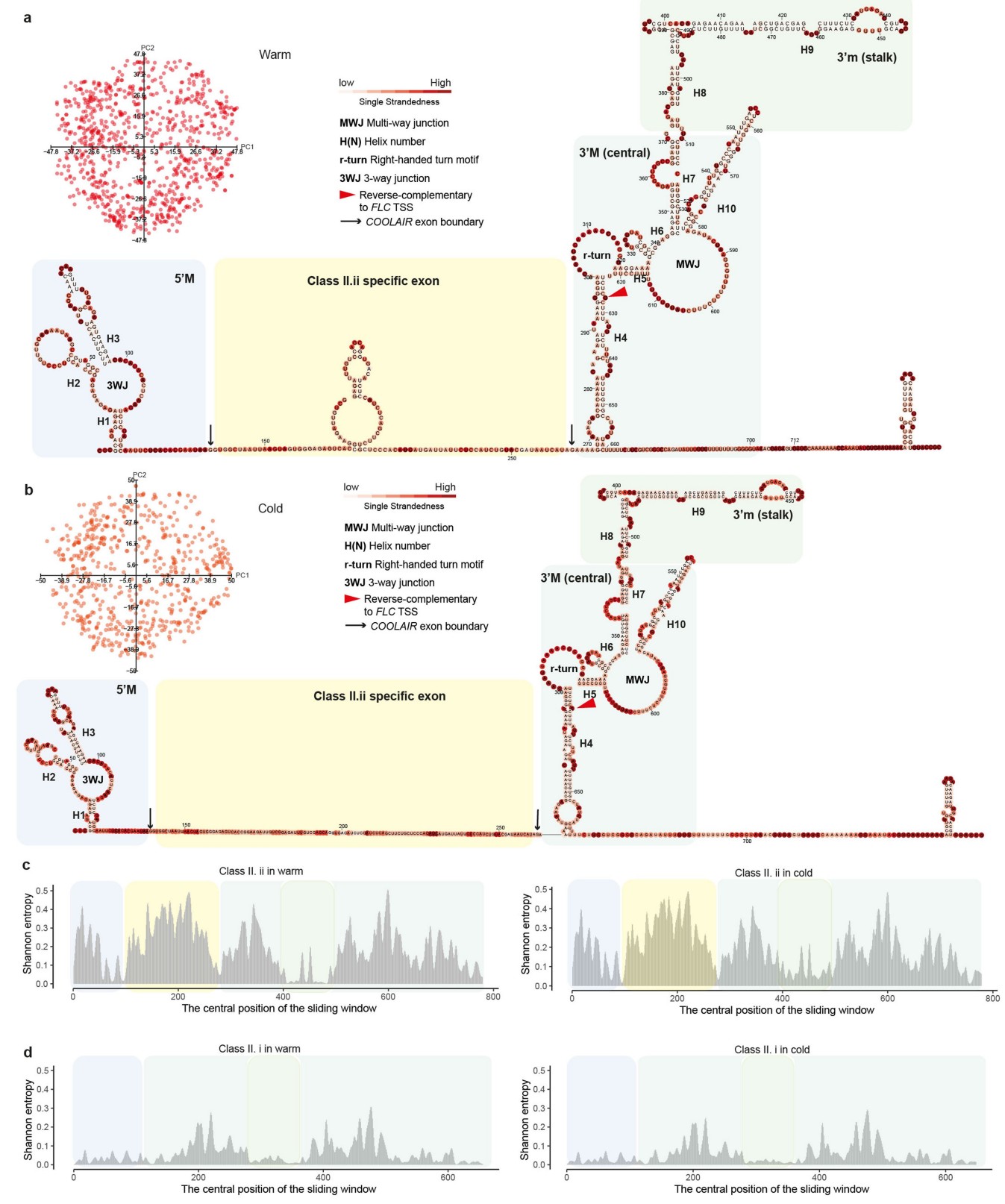

**Extended Data Fig. 6 | *COOLAIR* Class II.ii structure in warm and cold-grown plants. a**, The ensemble structure model of Class II.ii in warm-grown plants generated from CentroidFold (--engine CONTRAfold --sampling) was coloured by the likelihood of single-strandedness. The red triangle indicates the site corresponding to *FLC* TSS. The data was visualized using PCA. The four different coloured shadows refer to the four different domains. **b**, The

ensemble structure model of Class II.ii in cold-grown plants. The analysis was the same as in (**a**). **c**, The Shannon entropy of Class II.ii in warm and cold were calculated from (**a**) and (**b**) respectively. The shadows were coloured according to the domains in (**a**) and (**b**) respectively. **d**, The Shannon entropy of Class II.i in warm and cold were calculated from (Fig. 2) and (Fig. 3). The shadows were coloured according to the domains in (Fig. 2) and (Fig. 3) respectively.

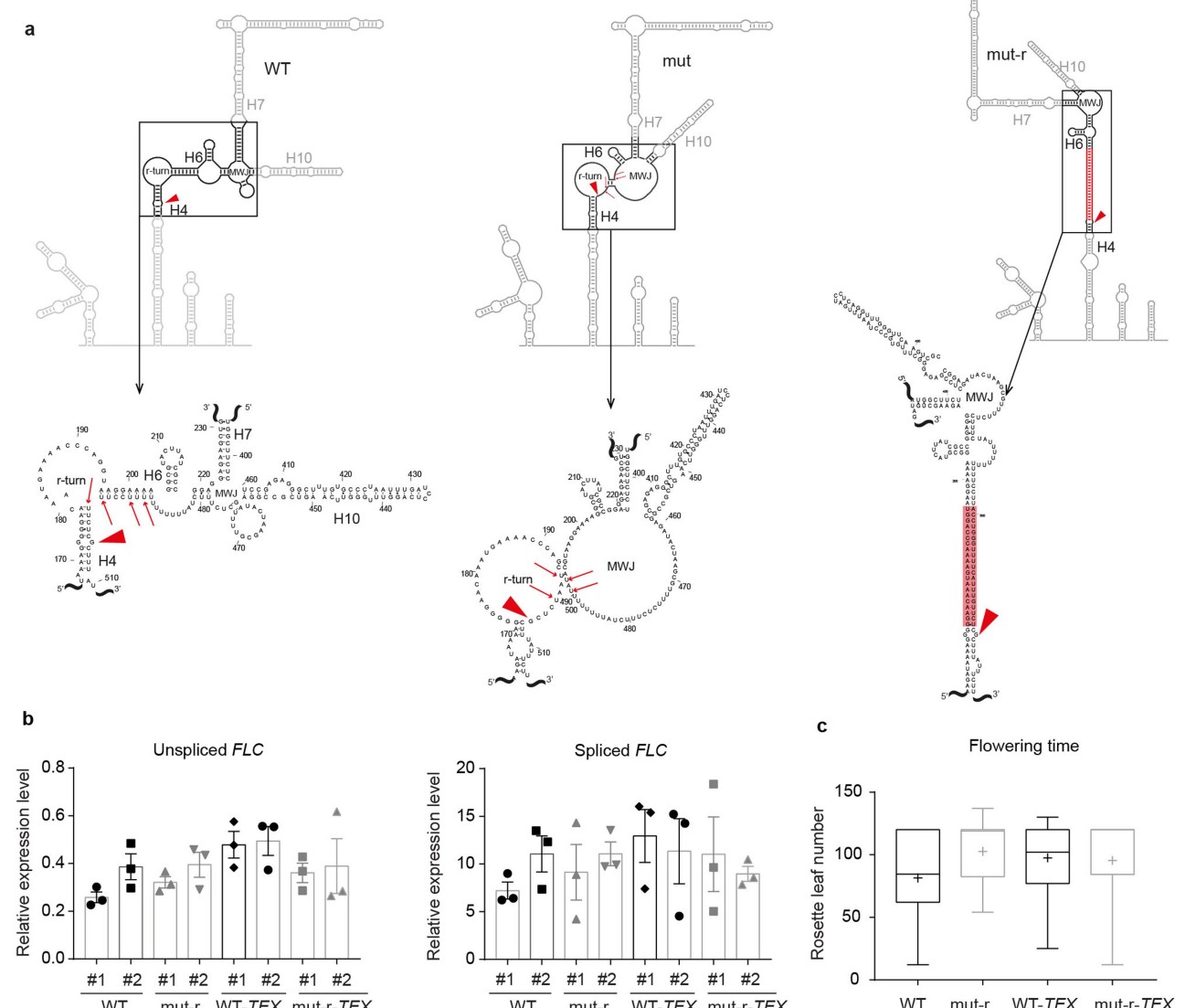

**Extended Data Fig. 7 | The effects of structural mutants mut-r on *COOLAIR* structure and *FLC* expression regulation. a**, The deduced schematic of the hyper-variable region in WT, mut and mut-r. The red triangles indicate the sites corresponding to *FLC* TSS. The mutation sites on the WT and mut were indicated by red arrows. The models are derived from (Fig. 4). The red shadow region in mut-r is the inserted sequence which increases the base-pairing in the H4–H6 region. **b**, The relative expression level of unspliced and spliced *FLC* transcript by RT-qPCR in the indicated genotypes. All RT-qPCR data are

presented as mean ± s.d.; n = 3 biologically independent experiments. The independent structural mutant transgenic lines are signified as #1 and #2. **c**, Box plots showing the flowering time of the indicated transgenic plants grown in warm conditions measured by rosette leaf numbers. Centre lines show median, box edges delineate 25th and 75th percentiles, bars extend to minimum and maximum values and '+' indicates the mean value. For each genotype population of mixed T3 lines are analyzed, from left to right, n = 36, 36, 35, and 36.

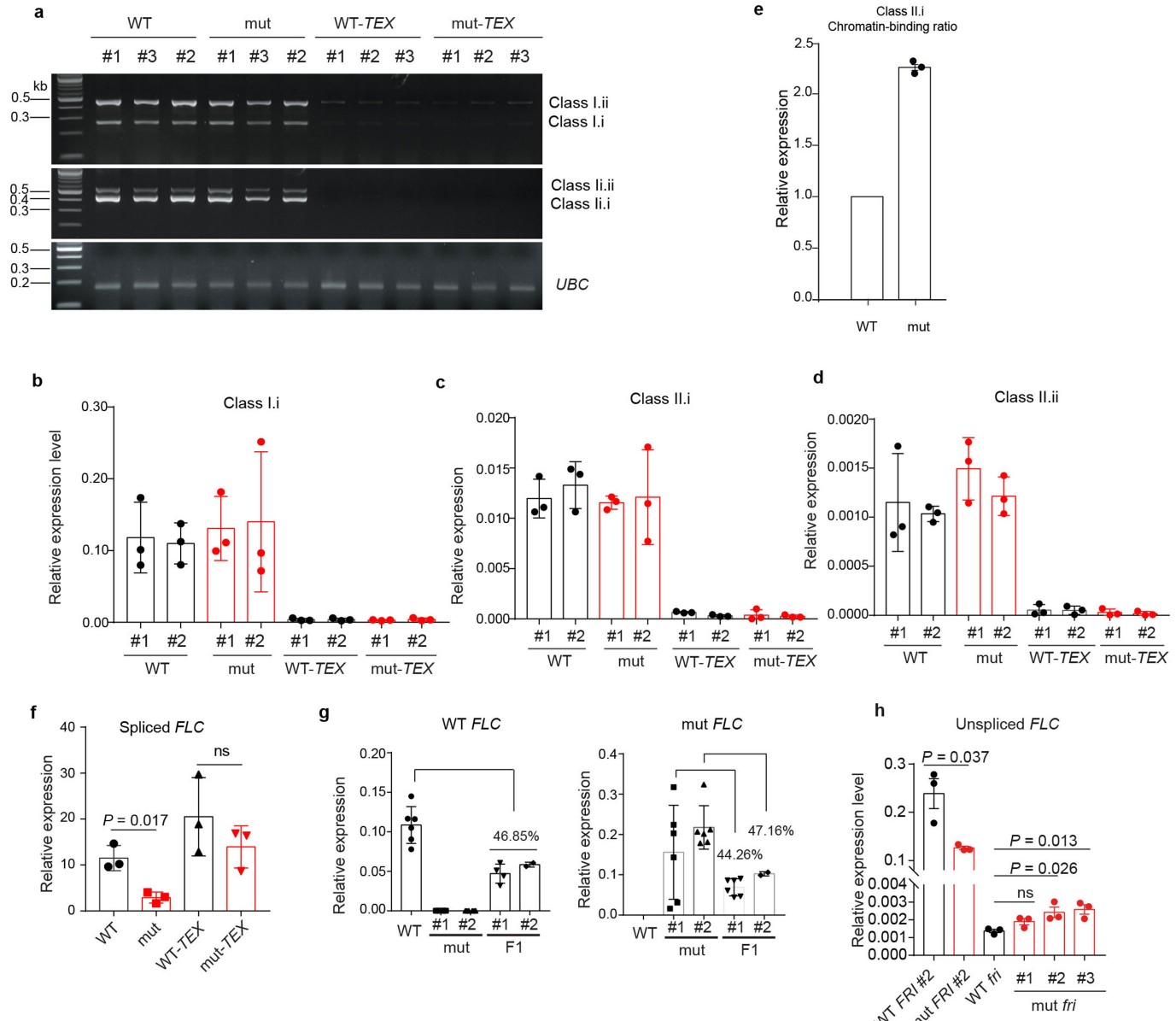

**Extended Data Fig. 8 | Interrogation of the effects of *COOLAIR* H4–H6 structurally hyper-variable region on *FLC* expression regulation. a**, RT-PCR of the spliced Class I and Class II *COOLAIR* isoforms in transgenic lines with and without the structural mutation, in both wild-type (mut and WT) or *TEX* backgrounds (mut-*TEX* and WT-*TEX*). *UBC* was used as control. 100 bp DNA ladder is shown on the left. For gel source data, see Supplementary Fig. 1. **b**–**d** and **f**, The relative expression level of spliced *COOLAIR* isoforms and *FLC* in the indicated genotypes assayed by RT-qPCR. Populations of mixed independent lines were analyzed for each genotype in (**f**). **e**, Chromatin-bound

proportion of Class II.i in mut line under warm conditions relative to WT, assayed by RT-qPCR. **g**, The relative expression level of the allele-specific *FLC* transcripts in F1 plants derived from the crossing between WT and structural mutant transgenic lines. **h**, The relative expression level of unspliced *FLC* transcript by RT-qPCR in WT and structural mutant in *FRI* background and loss of function (*fri*) background. One-way ANOVA with adjusted *P* value indicated in each comparison. All RT-qPCR data (**b**–**h**) are presented as mean ± s.d.; n = 3 biologically independent experiments. The independent structural mutant transgenic lines are signified as #1, #2 and #3.

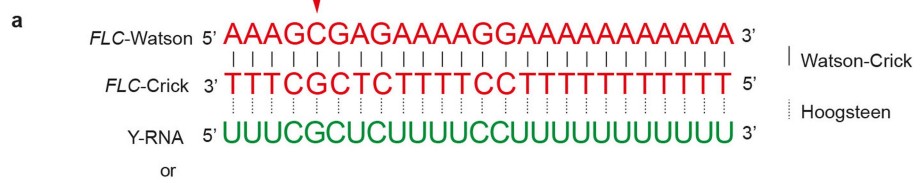

**a**

FLC-Watson 5' AAAGCGAGAAAAGGAAAAAAAAAA 3' | Watson-Crick

FLC-Crick 3' TTTCGCTCTTTTCCTTTTTTTTTTT 5' ⫶ Hoogsteen

Y-RNA 5' UUUCGCUCUUUUCCUUUUUUUUUU 3'

or

R-RNA 3' AAAGCGAGAAAAGGAAAAAAAAAA 5'

**b**

WT FLC 1 kb COOLAIR

**Negative Control**

DNA

DNA bw

RNA

RNA bw

overlay

ssDNA FLC dsDNA Y-RNA FLC dsDNA + Y-RNA

**FLC dsDNA around TSS**

DNA

DNA bw

RNA

RNA bw

overlay

ssDNA FLC dsDNA Y-RNA R-RNA FLC dsDNA + Y-RNA FLC dsDNA + R-RNA

**Positive Control**

DNA

DNA bw

RNA

RNA bw

overlay

En3 dsDNA PAPAS-RNA En3 dsDNA + PAPAS-RNA

**Extended Data Fig. 9 | Triplex formation tested by EMSA. a**, Potential triplex sequence content formed by pyrimidine Y-RNA or purine R-RNA within hyper-variable region. The Watson strand DNA and Crick strand DNA, shown in red, pair with each other via Watson-Crick bonds. The triplex-forming oligo RNA (Y-RNA or R-RNA) are shown in green and can bind *FLC* double-stranded DNA via Hoogsteen bonds. The Y-RNA corresponds to *COOLAIR* RNA with the same sequence content, while R-RNA corresponds to *FLC* RNA with the same sequence content. The red triangle indicates the site corresponding to *FLC* TSS. **b**, The panels show signal for DNA end-labelled with Cy5 (red colour) and RNA end-labelled with FAM (green colour). DNA/RNA bw, black and white

projection of the colour image; *FLC* dsDNA around TSS, Triplexator-predicted[45] triplex target site at *FLC* TSS within the hyper-variable region (corresponding to the red shadows in Extended Data Fig. 4e, f and Extended Data Fig. 5c); Negative control, oligonucleotide sequence upstream of *FLC* TSS (asterisks marks impurity in ssDNA oligo); Positive control, triplex-forming oligonucleotide sequence of human rDNA enhancer En3 with lncRNA *PAPAS*[21]. DNA-RNA triplex samples are shown in increasing ssRNA concentration with dsDNA:ssRNA ratios of 1:1, 1:2 and 1:4. For the positive control, the triplex sample is fixed at a ratio of 1:4. Data are representative of at least three independent experiments. For gel source data, see Supplementary Fig. 1.

# nature research

# Reporting Summary

Nature Research wishes to improve the reproducibility of the work that we publish. This form provides structure for consistency and transparency in reporting. For further information on Nature Research policies, see our Editorial Policies and the Editorial Policy Checklist.

## Statistics

For all statistical analyses, confirm that the following items are present in the figure legend, table legend, main text, or Methods section.

| n/a | Confirmed | |
|---|---|---|
| ☐ | ☒ | The exact sample size (*n*) for each experimental group/condition, given as a discrete number and unit of measurement |
| ☐ | ☒ | A statement on whether measurements were taken from distinct samples or whether the same sample was measured repeatedly |
| ☐ | ☒ | The statistical test(s) used AND whether they are one- or two-sided<br>*Only common tests should be described solely by name; describe more complex techniques in the Methods section.* |
| ☒ | ☐ | A description of all covariates tested |
| ☒ | ☐ | A description of any assumptions or corrections, such as tests of normality and adjustment for multiple comparisons |
| ☐ | ☒ | A full description of the statistical parameters including central tendency (e.g. means) or other basic estimates (e.g. regression coefficient) AND variation (e.g. standard deviation) or associated estimates of uncertainty (e.g. confidence intervals) |
| ☐ | ☒ | For null hypothesis testing, the test statistic (e.g. *F*, *t*, *r*) with confidence intervals, effect sizes, degrees of freedom and *P* value noted<br>*Give P values as exact values whenever suitable.* |
| ☒ | ☐ | For Bayesian analysis, information on the choice of priors and Markov chain Monte Carlo settings |
| ☒ | ☐ | For hierarchical and complex designs, identification of the appropriate level for tests and full reporting of outcomes |
| ☒ | ☐ | Estimates of effect sizes (e.g. Cohen's *d*, Pearson's *r*), indicating how they were calculated |

*Our web collection on statistics for biologists contains articles on many of the points above.*

## Software and code

Policy information about availability of computer code

| Data collection | All the sequencing data of COOLAIR and 18S rRNA were run on PacBio Sequel 3.0; The sequencing data of TenA, cspA and RRE61 were run on Illumina NextSeq 500; All QPCR reactions were run with LightCycler® 480; No software was used for other data; |
|---|---|
| Data analysis | For raw reads processing, ccs (https://github.com/PacificBiosciences/ccs) and Lima 1.11.0.<br>For sequence alignment, BLASR v5.3.3.<br>For RNA secondary structure analysis, ViennaRNA2.5.1, CONTRAFold 2.02 (http://contra.stanford.edu/contrafold/download.html) and CentroidFold 0.0.16 (https://github.com/satoken/centroid-rna-package).<br>For visualization of RNA secondary structure, VARNA v.3.93 and RNArtist (https://github.com/fjossinet/RNArtist).<br>For generating SHAPE mutation profile plots, R version 4.0.2.<br>For local base-pair probability analysis, R package astrochron v1.0.<br>For conformation analysis, Determination of the Variation of RNA structure conformation (DaVinci)(https://github.com/DingLab-RNAstructure/smStructure-seq) and Forgi 2.1.2.<br>For figure construction, Adobe Illustrator CC 2021.<br>For quantitative Real-Time PCR data analysis, Microsoft Excel for Mac v16.50. |

For manuscripts utilizing custom algorithms or software that are central to the research but not yet described in published literature, software must be made available to editors and reviewers. We strongly encourage code deposition in a community repository (e.g. GitHub). See the Nature Research guidelines for submitting code & software for further information.

## Data

Policy information about availability of data

All manuscripts must include a data availability statement. This statement should provide the following information, where applicable:

- Accession codes, unique identifiers, or web links for publicly available datasets
- A list of figures that have associated raw data
- A description of any restrictions on data availability

Sequencing data has been deposited in the Sequence Read Archive (SRA) (https://dataview.ncbi.nlm.nih.gov/object/PRJNA749291?reviewer=ql7br4e2j6n6vgovvp0cg3r5lh) under BioProject ID number PRJNA749291. A full list of DNA oligos, PCR primers and COOLAIR reference sequences is available in Supplementary Table 1. The raw data of RNA expression level, RT-PCR and ChIRP-qPCR that support the findings of this study are available in Source Data. Uncropped Images of EMSA and RT-PCR are available in Supplementary Figure 1. The TAIR (The Arabidopsis Information Resource) accession numbers for the genes analysed in this study are FLC (At5g10140) and COOLAIR (At5g01675). Standard reference genes EF1alpha (At5g60390), PP2A (At1g13320) and UBC (At5g25760) for gene expression were used for normalization.

# Field-specific reporting

Please select the one below that is the best fit for your research. If you are not sure, read the appropriate sections before making your selection.

☒ Life sciences  ☐ Behavioural & social sciences  ☐ Ecological, evolutionary & environmental sciences

For a reference copy of the document with all sections, see nature.com/documents/nr-reporting-summary-flat.pdf

# Life sciences study design

All studies must disclose on these points even when the disclosure is negative.

| | |
|---|---|
| Sample size | No statistical approach was used to predetermine sample size. Sample sizes were determined based on previous publications on similar previous experiments. The determined sample size was adequate as the differences between experimental groups was significant and reproducible.<br>RNA structure probing experiments (Yang, M. et al. Intact RNA structurome reveals mRNA structure-mediated regulation of miRNA cleavage in vivo. Nucleic Acids Res. 48, 8767–8781 (2020)).<br>RT-qPCR (Zhu, P., Lister, C. & Dean, C. Cold-induced Arabidopsis FRIGIDA nuclear condensates for FLC repression. Nature 599, 657–661 (2021).)<br>ChIRP-qPCR (Csorba, T., Questa, J. I., Sun, Q. & Dean, C. Antisense COOLAIR mediates the coordinated switching of chromatin states at FLC during vernalization. Proc. Natl. Acad. Sci. U. S. A. 111, 16160–16165 (2014).)<br>EMSA (Maldonado, R., Filarsky, M., Grummt, I. & Längst, G. Purine- and pyrimidine-triple-helix-forming oligonucleotides recognize qualitatively different target sites at the ribosomal DNA locus. RNA 24, 371–380 (2018).) |
| Data exclusions | No data was excluded from analysis. |
| Replication | The RNA structure experiments were performed with two independent biological replicates. The reproducibility of smStructure-seq has been confirmed by the positive control 18S rRNA (Pearson correlations of 0.95 with P value < 2.2e-16) that has been mentioned in Main Text. The reproducibility of RNA structural experiments has also been validated in the Rebuttal Fig. 4 (Supplementary Information of Peer Review Files). All key experimental findings in the structural mutation analysis were reproduced in more than three independent biological repeats with multiple technical replicates. Main conclusions were confirmed in different assays, including structural analysis in natural accession line, expression level analysis, ChIRP assay and genetic assays in transgenic COOLAIR structural mutants. |
| Randomization | Plants of different genotypes were grown side by side to minimize unexpected environmental variations during growth and experimentation. RNA structure probing was carried out in parallel on the same ThermoMixer (Eppendorf), with minimum covarying factors. Seedlings at the same developmental stage were collected and assessed randomly for each genotype/treatment. For ChIRP-qPCR, multiple seedlings were randomly collected from different plates for each replicate. For RNA expression, multiple, randomly selected individual plants were collected from a plate for each replicate. |
| Blinding | Blinding was not necessary for the molecular biology techniques, where bias could not be introduced as samples were treated together and identically. For bioimaging, the same settings were used for all comparisons. |

# Reporting for specific materials, systems and methods

We require information from authors about some types of materials, experimental systems and methods used in many studies. Here, indicate whether each material, system or method listed is relevant to your study. If you are not sure if a list item applies to your research, read the appropriate section before selecting a response.

## Materials & experimental systems

| n/a | Involved in the study |
|---|---|
| ☒ | ☐ Antibodies |
| ☒ | ☐ Eukaryotic cell lines |
| ☒ | ☐ Palaeontology and archaeology |
| ☒ | ☐ Animals and other organisms |
| ☒ | ☐ Human research participants |
| ☒ | ☐ Clinical data |
| ☒ | ☐ Dual use research of concern |

## Methods

| n/a | Involved in the study |
|---|---|
| ☒ | ☐ ChIP-seq |
| ☒ | ☐ Flow cytometry |
| ☒ | ☐ MRI-based neuroimaging |

