## [Peer Review File · Nature]

Manuscript Title: *In vivo* single-molecule analysis reveals COOLAIR RNA structural diversity

Editorial Notes:

Redactions – unpublished data

Reviewer Comments & Author Rebuttals

Reviewer Reports on the Initial Version:

Referees' comments:

Referee #1:

In the manuscript, Yang et al. developed a new technology, smStructure-seq, which combines chemical probing and long-read sequencing to determine the *in vivo* structures of individual transcript isoforms. Utilizing this technology, the authors identified different isoforms of COOLAIR and identified different conformational variants of a Class II.i isoform under cold and warm conditions. The authors further identified the structure of the H4/H5 hinge region on COOLAIR that directly interacts with FLC TSS to regulate FLC expression and flowering time.

These experiments highlight a potential “lncRNA thermometer” that controls flowering time, and make an important contribution to link RNA structure with environmental input to control gene expression. Nonetheless, the ideas of isoform identification and combining chemical probing and long-read sequencing are not entirely novel. The evidence that RNA secondary structure changes alter COOLAIR regulation of FLC locus is also incomplete. Although the authors identified different conformational variants of a Class II.i isoform under cold/warm conditions, how different conformational variants affect FLC TSS binding and their functional outcomes under cold/warm conditions is not characterized at all. In sum, more data are needed in order to support the authors' claims in the manuscript.

Major Comments

1. The authors developed smStructure-seq to determine the *in vivo* structures of individual transcript isoforms, which they claimed to have advantages over the PORE-cupine and DREEM. However, despite the obvious long-read advantage, the authors did not demonstrate the “higher accuracy” of smStructure-seq over those methods as claimed by the authors. This is particularly important since it is one of the key selling points of smStructure-seq mentioned by the authors. The fact that smStructure-seq still utilized DREEM to generate the RNA secondary structure models conflicts with the authors' claims that DREEM is limited by the Illumina-based platform (short-read sequencing). The authors should discuss and clarify this point and give DREEM more credit.

2. The authors' claim that smStructure-seq can reach single-molecule resolution is oversold. It actually generates the transcript conformation with a population averaging method, which is far from "single-molecule" resolution.

3. The benchmarking of smStructure-seq is not enough. Although the authors used 18S rRNA to demonstrate smStructure-seq can determine its secondary structure, the authors did not demonstrate smStructure-seq can determine different transcript isoforms accurately. The RNA duplex arch models are imputed but not directly proven. The authors can use psoralen crosslinking methods to prove the predicted RNA duplexes.

5. The overall low SHAPE reactivity is concerning. Although the authors can explain the low reactivity in Class I isoforms, the Class II isoforms also show low reactivity as well (the authors did not mention the exact percentage, but it seems low from the graph). This raises a concern of indicating incomplete SHAPE reaction or low modification efficiency in vivo.

6. The functional studies of conformation changes of the H4/H5 hinge region of COOLAIR are critical. In Fig. 4a, the authors design a mutant to disrupt RNA basepairing, which is argued to increase lncRNA-chromatin association and alter FLC silencing. The standard in RNA structure experiments is to make compensatory double mutants, and ask if the double mutant can restore RNA secondary structure and recover wild-type regulation.

7. The evidence that mutant COOLAIR allele is an RNA effect is incomplete. As the authors pointed out, the region of COOLAIR in question is also the FLC transcriptional start site on the other strand of DNA. The experiment in Fig. 4g to terminate COOLAIR transcription using the TEX allele showed that the mutant COOLAIR still had a significant effect on rosette number. This implies that the mutation is affecting FLC TSS on a DNA level rather than a true COOLAIR RNA effect. Similarly, the lack of trans effect in the F1 plants (ED Fig. 8g,h) is also readily explained by a cis DNA promoter effect.

Minor Comments

1. The expression level between Class I and II COOLAIR isoforms cannot be compared directly using the authors' approach because they were amplified using different primers sets. The level may be confounded by the priming efficiency of different primer sets.

2. Lines 112-113 and 157-158: The authors claimed Class II.ii has one conformation only, but no data was shown.

3. To really demonstrate that COOLAIR H4/H5 hinge binding is structure-dependent, the authors should perform a compensatory mutagenesis experiment - restoring the H4/H5 hinge from the four-nucleotide mutant with compensatory mutagenesis and check if it can restore FLC TSS binding.

4. To demonstrate the direct link between H4/H5 hinge-FLC TSS binding, the authors should perform some rescue experiments – blocking four-nucleotide mutant COOLAIR binding to FLC TSS and check if it can rescue the phenotype.

5. The way that the authors used "conformational changes" to describe a shift of the proportional levels among major and minor COOLAIR isoforms is misleading. The warm-major COOLAIR does not undergo "conformational changes" to form cold-minor from the same RNA molecule.

Referee #2:

This manuscript reports on experiments to investigate the structure of the COOLAIR lncRNA, a non-coding transcript produced from the Arabidopsis FLC locus that has been implicated in various aspects of FLC expression control.

The manuscript uses a method (SHAPE) that probes the structure of an RNA by modifying bases that are present in a single-stranded configuration. The authors find that the structure of COOLAIR isoforms differ in plants growing in the warm and cold, in particular a hinge region between helices 4 and 5 of the class II.i and II.ii isoforms, which corresponds to the transcription start site of the FLC gene. The hinge site was mutated to give a form predicted to enlarge the hinge region, with the mutated form showing a stronger association with FLC chromatin, reduced FLC transcript, and earlier flowering.

Although previous in vitro studies have shown that the COOLAIR structure is conserved and thus likely to be of functional significance, this study presents new in vivo data using a novel method to probe RNA structure. It will advance understanding of how lncRNAs can regulate gene expression as well as increase understanding of FLC regulation.

The methods used in the study are appropriate. The SHAPE method is a new development and was verified using the 18S rRNA. Whether a single rRNA is enough to show that the method is robust is one question that I would raise. It would have been good to see other RNAs where there is some knowledge of secondary structure used; other rRNAs, pre-miRNAs for example. This would give more confidence in the SHAPE method. Alternatively carrying out the same analysis on COOLAIR RNAs in other species could demonstrate that structures are conserved and also give additional evidence of functional significance. The other question I had over this method was whether it is possible to distinguish between RNA:RNA and RNA:DNA hybrids. On p. 4 the authors mention that Class I isoforms show few SHAPE modifications, which they suspect to be due to RNA:DNA hybrids. Although there are more SHAPE modifications in the Class II isoforms, how can the authors be sure the unmodified positions are all from RNA:RNA hybrids? In short I would like to have seen more demonstration that the novel SHAPE method generates robust data on RNA structure.

SHAPE analysis should be carried out on the plants carrying the mutant transgene to show that the predicted structures are present.

The presentation is generally good. It was a little difficult to follow the narrative about different transcript isoforms and structures at times. Some of the description of the detail of structure could be left out as it can be seen in the figures.

Referee #3:

The manuscript by Yang et al. titled "In vivo single-molecule RNA structure analysis reveals functionally important COOLAIR structural diversity" investigated structural diversity of Arabidopsis COOLAIR transcripts in response to temperature. COOLAIR represents a collection of antisense non-

coding transcripts derived from FLOWERING LOCUS C (FLC). By developing and employing in vivo single-molecule RNA structure profiling method (smStructure-seq), the authors revealed cold-induced structural variations in COOLAIR transcripts. One of the COOLAIR conformational variations that emerged in response to cold corresponds to the FLC transcription start site (TSS). Mutations altering this structural hinge resulted in compromised binding of COOLAIR to FLC chromatin, altered FLC expression and flowering time.

Many studies have demonstrated structural-functional relationships of various RNAs. These include but are not limited to in vivo and in vitro structural studies of mammalian Xist, fly roX1/2, and Arabidopsis microRNAs and rRNAs. It is becoming evident that RNA structural conformations are clearly deterministic in their interaction with cellular partners. With this manuscript, COOLAIR also joins the growing list of RNA structures to be studied in a native cellular environment. Overall, this manuscript is clearly written and most of the conclusions are supported by their experiments. However, the paper is somewhat lacking in novelty, and I do have few major and minor comments.

Major comments:

- The authors emphasized the development of the smStructure-seq method to study accurate and full-length RNA structures in vivo at single-molecule resolution. Tens of methods to study in vivo RNA structures using short-read sequencing exist. Yet these methods are not completely capable of determining structures of full-length RNA isoforms. Methods like PORE-cupine readily circumvented this problem and enabled structural profiling of full-length RNA isoforms using nanopore long-read sequencing. The current smStructure-seq presented in this manuscript simply replaces nanopore with the PacBio sequencing platform. smStructure-seq outperforms short-read-based methods in determining accurate and full-length RNA structures. However, the authors did not benchmark their smStructure-seq against other methods that utilize nanopore long-read sequencing coupled with machine learning approaches for base calling error correction. Even if smStructure-seq outperformed other long-read-based approaches in accuracy, this can be merely due to the high-fidelity nature of circular consensus-based PacBio sequencing. Therefore, I feel smStructure-seq simply utilizes advances in long-read sequencing and lacks any further novelty from a technical standpoint. To make this manuscript more impactful for one of the Nature family journals I suggest couple of experiments.

- RNA structures of COOLAIR transcripts derived from widely used Arabidopsis accessions Col-0 and northern Sweden Var2-6 have been resolved in vitro (Hawkes et al., 2016). It has also been shown that a single nucleotide polymorphism (SNP) in the H4 region of Var2-6 altered its structure and correlated this with increased FLC expression and delayed flowering time. In this study the authors showed the H4/H5 hinge region of COOLAIR Class II.i isoforms is required for its proper association to FLC chromatin at TSS. Introduction of four single-nucleotide mutations in the H4/H5 hinge region resulted in increased COOLAIR association to FLC chromatin, decreased FLC expression and early flowering phenotype. Using RNAfold, the authors predicted that these four mutations resulted in an enlarged and distorted structure of the H4/H5 hinge compared to wild type. In addition to using RNAfold prediction, I suggest that the authors validate the differential conformation of mutant (Col-0) and Var2-6 Class II.i COOLAIR transcripts in comparison to Col-0 wild type by performing smStructure-seq.

- It appears that introduction of mutation in Col-0 or the natural polymorphism of Var2-6 at the H4/H5 hinge is sufficient to change FLC expression and manifest a flowering time phenotype. Therefore, it would be interesting to investigate sequence divergence of H4/H5 regions across Arabidopsis natural accessions using SNP information in the 1001 genome project (Alonso-Blanco et al., 2016) and correlate with FLC expression (from Kawakatsu et al., 2016), temperature and structural variation (prediction-based). Ideally a couple of representative accessions from the above analysis should be experimentally validated for COOLAIR structures with smStructure-seq, FLC qRT-PCR and ChIRP assay. Together this could provide insights not only into functionally important COOLAIR in vivo structural conformations in response to temperature but also the significance of cis polymorphism in H4/H5 hinge region with regards to natural adaptation.

Minor suggestions:

- The authors should exemplify any differences and additional insights that smStructure-seq provides compared to in vitro probing of COOLAIR structures.
- Because there are enough data points (≥ 35) in Fig. 4g, a barplot (showing mean and standard deviation) of flowering time should be represented with box/violin plot (showing distribution). An appropriate statistical test should be performed as well. In addition, the authors could also consider showing a couple of representative images of actual flowering plants in extended data. This would help readers to visually appreciate how mutations in the structural hinge of Class II.i COOLAIR transcripts effects flowering time.
- The authors should mention the statistical test performed to call P-values in their Fig. 4f barplots showing FLC expression.
- Datasets under accession number PRJNA749291 should be made available for reviewers.

Author Rebuttals to Initial Comments:

Referee #1:

In the manuscript, Yang et al. developed a new technology, smStructure-seq, which combines chemical probing and long-read sequencing to determine the in vivo structures of individual transcript isoforms. Utilizing this technology, the authors identified different isoforms of COOLAIR and identified different conformational variants of a Class II.i isoform under cold and warm conditions. The authors further identified the structure of the H4/H5 hinge region on COOLAIR that directly interacts with FLC TSS to regulate FLC expression and flowering time.

These experiments highlight a potential “lncRNA thermometer” that controls flowering time and make an important contribution to link RNA structure with environmental input to control gene expression. Nonetheless, the ideas of isoform identification and combining chemical probing and long-read sequencing are not entirely novel. The evidence that RNA secondary structure changes alter COOLAIR regulation of FLC locus is also incomplete. Although the authors identified different conformational variants of a Class II.i isoform under cold/warm conditions, how different conformational variants affect FLC TSS binding and their functional outcomes under cold/warm conditions is not characterized at all. In sum, more data are needed in order to support the authors’ claims in the manuscript.

Major Comments

1. The authors developed smStructure-seq to determine the in vivo structures of individual transcript isoforms, which they claimed to have advantages over the PORE-cupine and DREEM. However, despite the obvious long-read advantage, the authors did not demonstrate the “higher accuracy” of smStructure-seq over those methods as claimed by the authors. This is particularly important since it is one of the key selling points of smStructure-seq mentioned by the authors. The fact that smStructure-seq still utilized DREEM to generate the RNA secondary structure models conflicts with the authors' claims that DREEM is limited by the Illumina-based platform (short-read sequencing). The authors should discuss and clarify this point and give DREEM more credit.

RESPONSE

We appreciate the referee’s knowledgeable comment. Building on the unique advantages of smStructure-seq, we have therefore developed a new structure analysis method, DaVinci, for smStructure-seq that facilitates the direct determination of RNA structure conformations at the single-molecule level.

Our current smStructure-seq has three significant advantages:

- 1) In dissecting the RNA isoform structure heterogeneity: The RNA structure information within the shared regions between isoforms cannot be distinguished by short read sequencing platforms (e.g., Illumina).
- 2) In determining the RNA structure information for single molecules: The nanopore-based method has an averaged error rate of 14% for both direct RNA and cDNA sequencing¹ so cannot give accurate single-molecule information. In contrast, PacBio platform can achieve 99.9% accuracy at the nucleotide level², thereby facilitating the accurate derivation of RNA structure for each single RNA molecule.
- 3) In dissecting the RNA structure conformation heterogeneity: the previous two computational approaches, DREEM³ and DRACO⁴ are chemical reactivity-based clustering methods. These methods tend to generate two mutation profiles with extremely high chemical modifications (more single-stranded RNA structure) or extremely low chemical modifications (more double-stranded RNA structure). These clusters directly reflect the similarity of chemical modification efficiencies, but do not directly represent the clusters of RNA structure conformations *per se*.

In addition to the advantage of high-accurate single-molecule sequencing, our smStructure-seq also incorporates a new analysis method that directly clusters the *in vivo* RNA structures derived from the mutation profile of each single molecule. This method named Determination of the Variation of RNA

structure conformation (DaVinci), incorporates the individual mutation profile (chemical modification profile) and derives the most-likely RNA structure conformation via a stochastic context-free grammar (SCFG) algorithm. Then the whole conformation space is identified and visualized via PCA analysis (detailed information in Methods section in the Main Text).

The DaVinci method, can accurately estimates RNA structure conformation at single-molecule resolution. To demonstrate the power of DaVinci, we performed our analysis pipeline on the HIV Rev response element (RRE) that has been reported to be able to adopt alternative conformations promoting different rates of virus replication⁵. DREEM used chemical reactivity-based clustering methods and identified two extreme conformations (conformation 1' and conformation 2' in Rebuttal Fig. 1a)³. However, DaVinci could identify at least three conformations (conformation 1, 2 and 3 in Rebuttal Fig. 1a) including an extra cryptic conformation⁶ (conformation 3) that cannot be identified by chemical reactivity-based clustering methods, e.g., DREEM³. This conformation (named as RRE61⁶) has been identified to have the ability to confer RevM10 resistance⁶.

A second example comes from analysis of primary-miRNAs RNA secondary structure, important as it affects the efficiency of Microprocessor and Dicing complexes⁷⁻⁹. Previous work¹⁰ has shown that a SWI2/SNF2 ATPase CHR2 can change pri-miRNA structures based on the ensemble structure model derived from average chemical modifications. In the *chr2-1* mutant, pri-miR319b has been showed to have a different ensemble structure from wild-type (Col0), potentially able to affect pri-miRNA processing from the terminal loop to the lower in *chr2-1*¹⁰. Using our DaVinci analysis pipeline, we revealed at least four conformations (Rebuttal Fig. 1b) in the conformation space of pri-miRNA319b in the Col0 and *chr2-1*. Rather than altering ensemble structures, CHR2 changes the proportion of specific individual conformations (Rebuttal Fig. 1b).

These results showed that DaVinci can identify the dynamic nature of *in vivo* RNA structure conformations for each single molecules, facilitating the investigation of the RNA structural conformation functionality *in vivo*. Therefore, our smStructure-seq is capable of generating single-RNA molecule structure for each RNA transcript (e.g., isoform).

We have described this method in Extended Data Fig. 2, Supplementary Discussion and Main Text line number 88-103.

Rebuttal Fig. 1 (Extended Data Fig. 2). **a**, The conformation landscape of HIV Rev response element (RRE) region. DaVinci was used to analyze the HIV RRE region³. The conformation 1' and conformation 2' were the conformation identified by DREEM³ which are close to the conformations identified by DaVinci (conformation 1 and conformation 2). A cryptic conformation⁶ with extended stem IV and short stem V has not been identified by DREEM³ but was shown by DaVinci (conformation 3). The black square is to highlight the conformation space around conformation 3. **b**, The conformation analysis on pri-miR319b¹⁰ by DaVinci. The PCA plot and the representative 76 schematic RNA structure models of pri-miR319b in Col0 and *chr2* mutant respectively.

2. The authors' claim that smStructure-seq can reach single-molecule resolution is oversold. It actually generates the transcript conformation with a population averaging method, which is far from "single-molecule" resolution.

RESPONSE

We thank the referee for this comment. The direct benefit of single-molecule reads in smStructure-seq is that we can accurately assign structure information to each *COOLAIR* isoform. Using our DaVinci pipeline, the smStructure-seq can identify each possible conformation with single-molecule resolution. smStructure-seq achieves this by deriving the RNA structure from each single mutation profile rather than averaged SHAPE mutation profiles, as discussed above.

3. The benchmarking of smStructure-seq is not enough. Although the authors used 18S rRNA to demonstrate smStructure-seq can determine its secondary structure, the authors did not demonstrate smStructure-seq can determine different transcript isoforms accurately. The RNA duplex arch models are imputed but not directly proven. The authors can use psoralen crosslinking methods to prove the predicted RNA duplexes.

RESPONSE

The intrinsic property of single-molecule detection of PacBio allows the accurate dissection of different transcript isoforms^{2,11-13}, since it requires no assembly step that attenuates the accuracy of isoform assignment. Therefore, smStructure-seq is intrinsically capable to determine different transcript isoforms accurately.

The long-distance RNA duplex (e.g., the H4 helix in the 3'M domain) forming the structural modules strongly agree with the evolutionary conserved structure¹⁴ (Rebuttal Fig. 5a, c), supporting the accuracy of our smStructure-seq. Additionally, psoralen crosslinking methods were based on Illumina-based methods, which could not achieve single-molecule resolution. The RNA duplex arch models were shown to highlight the distinct structural domains but to avoid confusion, we removed the arch models in the updated manuscript.

4. The overall low SHAPE reactivity is concerning. Although the authors can explain the low reactivity in Class I isoforms, the Class II isoforms also show low reactivity as well (the authors did not mention the exact percentage, but it seems low from the graph). This raises a concern of indicating incomplete SHAPE reaction or low modification efficiency *in vivo*.

RESPONSE

The value derived for 18S rRNA in Extended Data Fig. 1b supports the low SHAPE reactivity being due to the inaccessibility of base-paired nucleotides in the RNA structure, rather than incomplete SHAPE reaction or low modification efficiency *in vivo*. That 85.4% of the single-stranded nucleotides in the 18S rRNA were assigned with high SHAPE reactivity in Extended Data Fig. 1b, strongly supports that this level of SHAPE modification is sufficient for the RNA structure probing *in vivo* and our SHAPE reactivities represents high accuracy.

We interpret the low SHAPE reactivity in the Class I and Class II as related to the biological function. Our previous work has found that Class I *COOLAIR* forms stable R-loop structures at the 3' end of the *FLC* locus^{15,16}, this region spatially coincides with the lower accessibility *in vivo*. To further prove that the lower SHAPE accessibility of Class I is due to R-loop formation *in vivo*, we performed a DRIP-c (DNA-RNA immunoprecipitation followed by cDNA conversion) assay detecting *COOLAIR* RNA in RNA-DNA hybrids. The results showed that *COOLAIR* RNAs form stable RNA-DNA duplexes at the 3' end of *FLC* locus where the Class I *COOLAIR*s were mainly transcribed (Rebuttal Fig. 6a, b).

However, at the 5' end of the *FLC* locus (included in the transcription of Class II RNAs) there is no obvious *COOLAIR* RNA-DNA duplex formation. Therefore, altogether the low reactivity of the Class I isoforms is best interpreted as R-loop formation *in vivo*, whereas Class II can form RNA structures.

5. The functional studies of conformation changes of the H4/H5 hinge region of *COOLAIR* are critical. In Fig. 4a, the authors design a mutant to disrupt RNA basepairing, which is argued to increase *lncRNA*-chromatin association and alter *FLC* silencing. The standard in RNA structure experiments is to make compensatory double mutants and ask if the double mutant can restore RNA secondary structure and recover wild-type regulation.

RESPONSE

The mutant (mut line) we designed disrupted the RNA base-pairing of H4/H5 hinge within the H4-H6 region, by increasing the bulge and shortening H4 (Rebuttal Fig. 2a). This mutation resulted in increased *COOLAIR*-chromatin association (Fig. 4f), reduced *FLC* expression (Fig. 4g, Extended Data Fig. 6f) and early flowering (Fig. 4h, i). To add to this mutational analysis, we had generated a second mutant line (mut-r) reducing the bulge and extending H4 (Rebuttal Fig. 2a). This mutant achieved the *FLC* expression at very similar levels to wild type (Rebuttal Fig. 2b) and the flowering phenotype was recovered as similar as wild-type plants (Rebuttal Fig. 2c).

We have included this information in Extended Data Fig. 7 and Main Text line number 170-171.

Rebuttal Fig. 2 (Extended Data Fig. 7). a, The deduced schematic of the hyper-variable region in WT, mut and mut-r. The red triangles indicate the sites corresponding to the *FLC* TSS. The mutation sites on the WT and mut were indicated by red

arrows. The figures are derived from (Fig. 4). The red shadow region in mut-r is the inserted sequence which increases the base-pairing in the H4-H6 region. **b**, The relative expression level of unspliced and spliced *FLC* transcript by RT-qPCR in WT and structural mutant (mut-r) in different genotypes. All RT-qPCR data are presented as mean \pm s.d. The independent structural mutant transgenic lines are signified as #1 and #2. **c**, Box plots showing the flowering time of the indicated transgenic plants grown in warm conditions measured by rosette leaf number. Centre lines show median, box edges delineate 25th and 75th percentiles, bars extend to minimum and maximum values and '+' indicates the mean value. For each genotype population of mixed T3 lines are analyzed, from left to right, n=36, 36, 35, and 36.

6. The evidence that mutant *COOLAIR* allele is an RNA effect is incomplete. As the authors pointed out, the region of *COOLAIR* in question is also the *FLC* transcriptional start site on the other strand of DNA. The experiment in Fig. 4g to terminate *COOLAIR* transcription using the *TEX* allele showed that the mutant *COOLAIR* still had a significant effect on rosette number. This implies that the mutation is affecting *FLC* TSS on a DNA level rather than a true *COOLAIR* RNA effect. Similarly, the lack of trans effect in the F1 plants (ED Fig. 8g, h) is also readily explained by a cis DNA promoter effect.

RESPONSE

We thank the referee for pointing out this potential confusion. There is no statistical significance in unspliced and spliced *FLC* transcript level between WT-*TEX* and mut-*TEX* (Fig. 4g and Extended Data Fig. 6f) supporting the mut phenotype being a consequence of *COOLAIR*. That the flowering time of mut-*TEX* is slightly earlier than WT-*TEX* suggests more of an indirect role of *COOLAIR* at other flowering time loci. The mut-*TEX* rescues the mut phenotype (Fig. 4i) based on *FLC* expression again supporting the role that *COOLAIR* RNA has in this phenotype. Indeed, analysis of the F1 plants indicates the structural mutation only effects the local allele so we have edited the Main Text (line 182-184) to make this clearer.

Minor Comments

1. The expression level between Class I and II *COOLAIR* isoforms cannot be compared directly using the authors' approach because they were amplified using different primers sets. The level may be confounded by the priming efficiency of different primer sets.

RESPONSE

The pie-chart has been removed from the plot and the relative percentage within each Class has been listed in a table in Extended Data Fig. 1a.

2. Lines 112-113 and 157-158: The authors claimed Class II.ii has one conformation only, but no data was shown.

RESPONSE

The data was added in the Extended Data Fig. 5 and the Main Text is updated accordingly (line 143-144).

3. To really demonstrate that *COOLAIR* H4/H5 hinge binding is structure-dependent, the authors should perform a compensatory mutagenesis experiment - restoring the H4/H5 hinge from the four-nucleotide mutant with compensatory mutagenesis and check if it can restore *FLC* TSS binding.

RESPONSE

An additional mutagenesis experiment has been performed by insertion of a fragment reducing the bulge and extending H4 and the discussion is detailed in Major comment #5. Briefly, this new mutant (mut-r) achieved *FLC* expression and flowering time at similar levels to wild type (Rebuttal Fig. 2b, c).

To check the chromatin association in the mutants, we also performed the ChIRP experiments and the results showed that the ChIRP signal in this mut-r was similar to wild type and even slightly reduced at the *FLC* TSS region (Rebuttal Fig. 3b).

Rebuttal Fig. 3. **a**, Enrichment of Class II RNA by ChIRP probes detected by RT-qPCR in WT and mut-r line. Class I and *UBC* RNAs were used as negative controls. RNase+, RNase A/T1 mix was added during the hybridization. **b**, ChIRP experiments showing the DNA enrichment at *FLC* TSS region mediated by Class II *COOLAIR*, detected by qPCR Mean \pm s.e.m, n=3. The position of “0” indicates the position of *FLC* TSS.

4. To demonstrate the direct link between H4/H5 hinge-*FLC* TSS binding, the authors should perform some rescue experiments – blocking four-nucleotide mutant *COOLAIR* binding to *FLC* TSS and check if it can rescue the phenotype.

RESPONSE

We have added extra experiments (discussed in *Major comment #5* and *Minor comment #3*) and edited the discussion in Main Text to make this clearer (line 195-200).

5. The way that the authors used “conformational changes” to describe a shift of the proportional levels among major and minor *COOLAIR* isoforms is misleading. The warm-major *COOLAIR* does not undergo “conformational changes” to form cold-minor from the same RNA molecule.

RESPONSE

Thank you, we have edited the Main Text to correct this, lines 136-138.

Referee #2:

This manuscript reports on experiments to investigate the structure of the COOLAIR lncRNA, a non-coding transcript produced from the Arabidopsis FLC locus that has been implicated in various aspects of FLC expression control.

The manuscript uses a method (SHAPE) that probes the structure of an RNA by modifying bases that are present in a single-stranded configuration. The authors find that the structure of COOLAIR isoforms differ in plants growing in the warm and cold, in particular a hinge region between helices 4 and 5 of the class II.i and II.ii isoforms, which corresponds to the transcription start site of the FLC gene. The hinge site was mutated to give a form predicted to enlarge the hinge region, with the mutated form showing a stronger association with FLC chromatin, reduced FLC transcript, and earlier flowering.

Although previous in vitro studies have shown that the COOLAIR structure is conserved and thus likely to be of functional significance, this study presents new in vivo data using a novel method to probe RNA

structure. It will advance understanding of how lncRNAs can regulate gene expression as well as increase understanding of FLC regulation.

1. The methods used in the study are appropriate. The SHAPE method is a new development and was verified using the 18S rRNA. Whether a single rRNA is enough to show that the method is robust is one question that I would raise. It would have been good to see other RNAs where there is some knowledge of secondary structure used; other rRNAs, pre-miRNAs for example. This would give more confidence in the SHAPE method. Alternatively carrying out the same analysis on COOLAIR RNAs in other species could demonstrate that structures are conserved and also give additional evidence of functional significance.

RESPONSE

We thank the referee for this suggestion. The SHAPE chemical probing is well-established and widely used to probe RNA structure^{17–22}. And PacBio can achieve 99.9% accuracy at the nucleotide level², facilitating the single-molecule accuracy. Thus, we consider it is not necessary to prove the probing accuracy of smStructure-seq using other RNAs.

Besides, 18S rRNA, seen as the gold standard in several RNA structure probing experiments^{18,23} has long sequence, diverse RNA structure motifs and well-defined consensus structure. These features make it the best candidate to test long-read single-molecule RNA probing experiments, e.g., smStructure-seq. To further confirm the accuracy and reproducibility of smStructure-seq, we also performed smStructure-seq on 18S rRNA in a different genotype (Var2-6) under different growing conditions and we found that smStructure-seq is highly reproducible (Rebuttal Fig. 4).

Rebuttal Fig. 4. The Pearson correlation of SHAPE reactivity of 18S rRNA between Columbia and Var2-6 genotypes under different growing condition.

The advantages of smStructure-seq are that it simultaneously informs on: 1) RNA isoform structure heterogeneity; 2) RNA structure information for single molecules and 3) RNA structure conformation heterogeneity. These advanced features can facilitate the functional dissection of lncRNA, e.g., COOLAIRs. To demonstrate the advantage of smStructure-seq, we also identified the *in vivo* RNA structure of a rare Class II isoform (Class II.iv) in Var2-6 near isogenic line (NIL) in ColFRI background (Rebuttal Fig. 5d and Rebuttal Fig. 8).

Class II COOLAIR has a conserved architecture in *Brassicaceae* species¹⁴ (Rebuttal Fig. 5a, derived from¹⁴). However, different species show some variation in local structures. For example, *B. rapa*¹⁴ has

been shown to have a variable multiple-way-junction loop around H4-H6 regions (Rebuttal Fig. 5b, derived from¹⁴), while other structural features including the domain arrangement (i.e., 5'M, 3'M and 3'm), helix topology (e.g., conserved helix position and structures of H1, H2 and H3, et.al.) are conserved. Comparing the isoform Class II.iv in Var2-6 NIL (Rebuttal Fig. 5d) with consensus secondary structures (Rebuttal Fig. 5a), *B. rapa* (Rebuttal Fig. 5b) and Class II.i in ColFRI (Rebuttal Fig. 5c), the conserved three-domain arrangement and conserved helix (H1-H10) topology agrees well. Only the H4-H6 region adopts different conformations in different species and isoforms. The mutations (mut line) altering this region (Fig. 4) show it is indeed functionally important for *COOLAIR* to regulate *FLC* expression *in vivo*.

The conserved domain arrangement and helix topology of RNA structure captured by smStructure further supports the efficacy and advantages of our method to probe the functional important RNA structure *in vivo*.

We have added these data in Extended Data Fig. 4 and the description in the Main Text (line 113-120).

Rebuttal Fig. 5. **a**, Consensus diagram shows conservation of the secondary structure of Class II *COOLAIR*s, where coloured boxes represent the percentage of conservation across species. The figure is derived from our previous work¹⁴. **b**, *B. rapa* *COOLAIR* sequence can form conserved RNA secondary structure and validated by 3S in our previous work¹⁴. **c**, The dominant conformation (Warm-Conformation 1 from Fig. 1a) was similar to consensus diagram. **d**, The *in vivo* Class II.iv RNA structure (from Var2-6) was derived from smStructure-seq. The truncated line between 5'M and 3'M were indicated single stranded region and ignored here for the comparison with (a). The full structure model of Class II.iv is in Rebuttal Fig. 8.

2. The other question I had over this method was whether it is possible to distinguish between RNA:RNA and RNA:DNA hybrids. On p. 4 the authors mention that Class I isoforms show few SHAPE modifications, which they suspect to be due to RNA:DNA hybrids. Although there are more SHAPE

modifications in the Class II isoforms, how can the authors be sure the unmodified positions are all from RNA:RNA hybrids? In short, I would like to have seen more demonstration that the novel SHAPE method generates robust data on RNA structure.

RESPONSE

We interpret the low SHAPE reactivity in the Class I and Class II as related to the biological function. Our previous work has found that Class I *COOLAIR* forms stable R-loop structures at the 3' end of the *FLC* locus^{15,16}, this region spatially coincides with the lower accessibility *in vivo*. To further prove that the lower SHAPE accessibility of Class I is due to R-loop formation *in vivo*, we performed a DRIP-c (DNA-RNA immunoprecipitation followed by cDNA conversion) assay detecting *COOLAIR* RNA in RNA-DNA hybrids. The results showed that *COOLAIR* RNAs form stable RNA-DNA duplexes at the 3' end of *FLC* locus where the Class I *COOLAIR*s were mainly transcribed (Rebuttal Fig. 6a, b). However, at the 5' end of the *FLC* locus (included in the transcription of Class II RNAs) there is no obvious *COOLAIR* RNA-DNA duplex formation.

Therefore, altogether the low reactivity of the Class I isoforms is best interpreted as R-loop formation *in vivo*, whereas Class II can form RNA structures.

[REDACTED]

3. *SHAPE analysis should be carried out on the plants carrying the mutant transgene to show that the predicted structures are present.*

RESPONSE

We thank the referee for this suggestion. We carried out the RNA structure probing on the mutant (mut) line (Rebuttal Fig. 7a, b). Our smStructure-seq analysis (DaVinci) showed that there is at least one dominant conformation in the whole conformation space (Rebuttal Fig. 7d) and the RNA structure confirmed that the H4/H5 hinge within H4-H6 region was enlarged in the mutant (Rebuttal Fig. 7d).

We have added these data in Fig. 4b-d and the description in the Main Text (lines 159-160).

Rebuttal Fig. 7 (Fig. 4). **a**, Schematic illustration of *FLC* and *COOLAIR* in the wild type. Untranslated regions (UTRs) are shown in grey boxes and exons in black boxes. The mutation site is indicated by a red arrow. 1kb, 1 kilobase. The red triangle indicates the site corresponding to the *FLC* TSS. **b**, The schematic of the mutation on the major conformation structure (Warm-Conformation 1) is derived from Fig. 2a. **c**, The H4-H6 hyper-variable region in wildtype Class II.i from Fig. 2a. The red triangle indicates the site corresponding to the *FLC* TSS. The mutation sites are indicated by red arrows. **d**, The H4-H6 hyper-variable region of Class II.i in mutant line (mut). The mutation sites are indicated by red arrows and are at the same position as in the (b, c). Visualization of *in vivo* structural conformations of Class II.i in warm-grown mut plants is showed in the inset. Each dot represents a unique single structure derived from each single-molecule mutation profile. All the dots were directly generated from around 300 individual raw mutation profiles. Data were visualized using Principal Component Analysis (PCA).

4. The presentation is generally good. It was a little difficult to follow the narrative about different transcript isoforms and structures at times. Some of the description of the detail of structure could be left out as it can be seen in the figures.

RESPONSE

We have followed the referee's suggestion and the manuscript has been updated accordingly. Generally, I added more illustrations of gene structure of *COOLAIR* isoforms to assist the readability, for example, in Fig. 1, Fig. 4 and Extended Data Fig. 4a. The trivial descriptions on the details of structure have been removed.

Referee #3:

The manuscript by Yang et al. titled “In vivo single-molecule RNA structure analysis reveals functionally important COOLAIR structural diversity” investigated structural diversity of *Arabidopsis* COOLAIR transcripts in response to temperature. COOLAIR represents a collection of antisense non-coding transcripts derived from FLOWERING LOCUS C (FLC). By developing and employing in vivo single-molecule RNA structure profiling method (smStructure-seq), the authors revealed cold-induced structural variations in COOLAIR transcripts. One of the COOLAIR conformational variations that emerged in response to cold corresponds to the FLC transcription start site (TSS). Mutations altering this structural hinge resulted in compromised binding of COOLAIR to FLC chromatin, altered FLC expression and flowering time.

Many studies have demonstrated structural-functional relationships of various RNAs. These include but are not limited to in vivo and in vitro structural studies of mammalian Xist, fly roX1/2, and *Arabidopsis* microRNAs and rRNAs. It is becoming evident that RNA structural conformations are clearly deterministic in their interaction with cellular partners. With this manuscript, COOLAIR also joins the growing list of RNA structures to be studied in a native cellular environment. Overall, this manuscript is clearly written and most of the conclusions are supported by their experiments. However, the paper is somewhat lacking in novelty, and I do have few major and minor comments.

Major comments:

1. The authors emphasized the development of the smStructure-seq method to study accurate and full-length RNA structures in vivo at single-molecule resolution. Tens of methods to study in vivo RNA structures using short-read sequencing exist. Yet these methods are not completely capable of determining structures of full-length RNA isoforms. Methods like PORE-cupine readily circumvented this problem and enabled structural profiling of full-length RNA isoforms using nanopore long-read sequencing. The current smStructure-seq presented in this manuscript simply replaces nanopore with the PacBio sequencing platform. smStructure-seq outperforms short-read-based methods in determining accurate and full-length RNA structures. However, the authors did not benchmark their smStructure-seq against other methods that utilize nanopore long-read sequencing coupled with machine learning approaches for base calling error correction. Even if smStructure-seq outperformed other long-read-based approaches in accuracy, this can be merely due to the high-fidelity nature of circular consensus-based PacBio sequencing. Therefore, I feel smStructure-seq simply utilizes advances in long-read sequencing and lacks any further novelty from a technical standpoint. To make this manuscript more impactful for one of the Nature family journals I suggest couple of experiments.

RESPONSE

The smStructure-seq provides so much more than just utilise PacBio sequencing technology.

Our current smStructure-seq has three significant advantages:

- 1) In dissecting the RNA isoform structure heterogeneity: The RNA structure information within the shared regions between isoforms cannot be distinguished by short read sequencing platforms (e.g., Illumina).
- 2) In determining the RNA structure information for single molecules: The nanopore-based method has an averaged error rate of 14% for both direct RNA and cDNA sequencing¹ so cannot give accurate single-molecule information. In contrast, PacBio platform can achieve 99.9% accuracy at the nucleotide level², facilitating the accurate derivation of RNA structure for each single RNA molecule.
- 3) In dissecting the RNA structure conformation heterogeneity: the previous two computational approaches, DREEM³ and DRACO⁴ are chemical reactivity-based clustering methods. These methods tend to generate two mutation profiles with extremely high chemical modifications (more single-stranded RNA structure) or extremely low chemical modifications (more double-stranded RNA structure). These clusters directly reflect the similarity of chemical modification efficiencies, but do not directly represent the clusters of RNA structure conformations *per se*.

In addition to the advantage of high-accurate single-molecule sequencing, our smStructure-seq also incorporates a new analysis method that directly clusters the *in vivo* RNA structures derived from the mutation profile of each single molecule. This new method (**D**etermination of the **V**ariation of RNA structure **c**onformation, i.e., DaVinci) incorporates the individual mutation profile (chemical modification profile) and derives the most-likely RNA structure conformation for each single molecule via a stochastic context-free grammar (SCFG) algorithm. The detailed validation and discussion were in *Major Comments #1* of Referee 1.

Thus, smStructure-seq can identify the dynamic structure conformation for each isoform at single-molecule resolution. The full discussion is included in **Extended Data Fig. 2, Supplementary Discussion and Main Text line number 88-103**.

2. *RNA structures of COOLAIR transcripts derived from widely used Arabidopsis accessions Col-0 and northern Sweden Var2-6 have been resolved in vitro (Hawkes et al., 2016). It has also been shown that a single nucleotide polymorphism (SNP) in the H4 region of Var2-6 altered its structure and correlated this with increased FLC expression and delayed flowering time. In this study the authors showed the H4/H5 hinge region of COOLAIR Class II.i isoforms is required for its proper association to FLC chromatin at TSS. Introduction of four single-nucleotide mutations in the H4/H5 hinge region resulted in increased COOLAIR association to FLC chromatin, decreased FLC expression and early flowering phenotype. Using RNAfold, the authors predicted that these four mutations resulted in an enlarged and distorted structure of the H4/H5 hinge compared to wild type. In addition to using RNAfold prediction, I suggest that the authors validate the differential conformation of mutant (Col-0) and Var2-6 Class II.i COOLAIR transcripts in comparison to Col-0 wild type by performing smStructure-seq.*

RESPONSE

We thank the referee for this suggestion. We have performed RNA structure probing on mutation line (mut) by smStructure-seq (**Rebuttal Fig. 7**). Our smStructure-seq analysis (DaVinci) showed that there is at least one dominant conformation in the whole conformation space (**Rebuttal Fig. 7d**) and the RNA structure confirmed that the H4/H5 hinge within H4-H6 region was enlarged in the mutant (**Rebuttal Fig. 7d**). We have added these data in **Fig. 4b-d** and the description in the Main Text (**lines 159-160**).

We also performed smStructure-seq on a genotype carrying the Var2-6 *FLC* allele introgressed into *ColFRI* (**Rebuttal Fig. 8a**). This experiment showed the *in vivo* structure of Class II.iv has a very short H4 and a merged H5 to extend H6 (**Rebuttal Fig. 8a, b**). These structural changes occur in the region complementary to the *FLC* TSS (**Rebuttal Fig. 8b**), i.e., hyper-variable region. We have added these data in **Extended Data Fig. 4** and the description in the Main Text (**line 113-120**).

Rebuttal Fig. 8 (Extended Data Fig. 4). **a**, Schematic illustration of *FLC* and *COOLAIR* gene structure. Untranslated regions (UTRs) are shown in grey boxes and exons in black boxes. The extra exon of Class II.iv in Var2-6²⁴ is indicated by a green line. 1kb, 1 kilobase. The red triangle indicates the site corresponding to the *FLC* TSS. **b**, The *in vivo* structure of Class II.iv in Var2-6 line. **c**, The schematic RNA structure model of Warm-Conformation 1 is from (Fig. 4b) and Class II.iv from (b). The hyper-variable regions are showed in the black-square.

3. It appears that introduction of mutation in *Col-0* or the natural polymorphism of *Var2-6* at the H4/H5 hinge is sufficient to change *FLC* expression and manifest a flowering time phenotype. Therefore, it would be interesting to investigate sequence divergence of H4/H5 regions across *Arabidopsis* natural accessions using SNP information in the 1001 genome project (Alonso-Blanco et al., 2016) and correlate with *FLC* expression (from Kawakatsu et al., 2016), temperature and structural variation (prediction-based). Ideally a couple of representative accessions from the above analysis should be experimentally validated for *COOLAIR* structures with *smStructure-seq*, *FLC* qRT-PCR and ChIRP assay. Together this could provide insights not only into functionally important *COOLAIR* *in vivo* structural conformations in response to temperature but also the significance of *cis* polymorphism in H4/H5 hinge region with regards to natural adaptation.

RESPONSE

We thank referee for this suggestion. However, as well as SNPs across the *FLC* locus, *FLC* expression is affected by other sequence variations at other loci, e.g., *FRIGIDA*. Therefore, using the 1001 genome project to find the correlation between sequence divergence of H4-H6 regions and *FLC* expression level will oversimplify the relationship between *COOLAIR* and *FLC* regulation.

Besides, as manifested in the Var2-6, the polymorphisms in the *FLC* locus may change the splicing pattern and generate new *COOLAIR* isoforms. And our recent work²⁵ has shown different isoforms have distinct functions in *FLC* regulation under different temperature conditions. Therefore, the function of structural conformation of *COOLAIR* in *FLC* regulation in natural accessions or in response to temperature is more complicated than just correlation with the expression level.

Minor suggestions:

1, *The authors should exemplify any differences and additional insights that smStructure-seq provides compared to in vitro probing of COOLAIR structures.*

RESPONSE

We thank the referee for this suggestion. We have added these data in **Extended Data Fig. 3** and the increased description in the Main Text (**line 108-112**).

2, *Because there are enough data points (≥ 35) in Fig. 4g, a barplot (showing mean and standard deviation) of flowering time should be represented with box/violin plot (showing distribution). An appropriate statistical test should be performed as well. In addition, the authors could also consider showing a couple of representative images of actual flowering plants in extended data. This would help readers to visually appreciate how mutations in the structural hinge of Class II.i COOLDAIR transcripts effects flowering time.*

RESPONSE

We thank the referee for this suggestion. The bar plot has been updated to box plot (**Fig. 4i** in manuscript) and the representative images (**Rebuttal Fig. 9**) of actual flowering plants have added to the Main Text (**Fig. 4h**).

Rebuttal Fig. 9 (Fig. 4h). Photographs showing flowering phenotype of WT and mut plants after cold exposure. Scale bar, 50 mm.

3, *The authors should mention the statistical test performed to call P-values in their Fig. 4f barplots showing FLC expression.*

RESPONSE

We thank for referee's suggestion. And the *P* values has been updated in the figure legend.

4, Datasets under accession number PRJNA749291 should be made available for reviewers.

RESPONSE

These are made available through the link

<https://dataview.ncbi.nlm.nih.gov/object/PRJNA749291?reviewer=ql7br4e2j6n6vgovvp0cg3r5lh>.

References

1. Workman, R. E. *et al.* Nanopore native RNA sequencing of a human poly(A) transcriptome. *Nat. Methods* **16**, 1297–1305 (2019).
2. Wenger, A. M. *et al.* Accurate circular consensus long-read sequencing improves variant detection and assembly of a human genome. *Nat. Biotechnol.* **37**, 1155–1162 (2019).
3. Tomezsko, P. J. *et al.* Determination of RNA structural diversity and its role in HIV-1 RNA splicing. *Nature* **582**, 438–442 (2020).
4. Morandi, E. *et al.* Genome-scale deconvolution of RNA structure ensembles. *Nat. Methods* **18**, 249–252 (2021).
5. Sherpa, C., Rausch, J. W., Le Grice, S. F. J., Hammarskjold, M. L. & Rekosh, D. The HIV-1 Rev response element (RRE) adopts alternative conformations that promote different rates of virus replication. *Nucleic Acids Res.* **43**, 4676–4686 (2015).
6. Legiewicz, M. *et al.* Resistance to RevM10 inhibition reflects a conformational switch in the HIV-1 Rev response element. *Proc. Natl. Acad. Sci. U. S. A.* **105**, 14365–14370 (2008).
7. Song, L., Axtell, M. J. & Fedoroff, N. V. RNA secondary structural determinants of miRNA precursor processing in *Arabidopsis*. *Curr. Biol.* **20**, 37–41 (2010).
8. Werner, S., Wollmann, H., Schneeberger, K. & Weigel, D. Structure determinants for accurate processing of miR172a in *Arabidopsis thaliana*. *Curr. Biol.* **20**, 42–48 (2010).
9. Mateos, J. L., Bologna, N. G., Chorostecki, U. & Palatnik, J. F. Identification of microRNA processing determinants by random mutagenesis of *Arabidopsis* MIR172a precursor. *Curr. Biol.* **20**, 49–54 (2010).
10. Wang, Z. *et al.* SWI2/SNF2 ATPase CHR2 remodels pri-miRNAs via Serrate to impede miRNA production. *Nature* **557**, 516–521 (2018).
11. Zhao, L. *et al.* Analysis of transcriptome and epitranscriptome in plants using PacBio Iso-Seq and Nanopore-based direct RNA sequencing. *Front. Genet.* **10**, 1–14 (2019).
12. An, D., Cao, H., Li, C., Humbeck, K. & Wang, W. Isoform sequencing and state-of-art applications for unravelling complexity of plant transcriptomes. *Genes (Basel)*. **9**, 43 (2018).
13. Mays, A. D. *et al.* Single-molecule real-time (SMRT) full-length RNA-sequencing reveals novel and distinct mRNA isoforms in human bone marrow cell subpopulations. *Genes (Basel)*. **10**, 253 (2019).
14. Hawkes, E. J. *et al.* COOLAIR antisense RNAs form evolutionarily conserved elaborate secondary structures. *Cell Rep.* **16**, 3087–3096 (2016).
15. Sun, Q., Csorba, T., Skourti-Stathaki, K., Proudfoot, N. J. & Dean, C. R-Loop stabilization represses antisense transcription at the *Arabidopsis* *FLC* locus. *Science (80-.)*. **340**, 619–621 (2013).
16. Xu, C. *et al.* R-loop resolution promotes co-transcriptional chromatin silencing. *Nat. Commun.* **12**, 1790 (2021).

17. Merino, E. J., Wilkinson, K. A., Coughlan, J. L. & Weeks, K. M. RNA structure analysis at single nucleotide resolution by selective 2'-hydroxyl acylation and primer extension (SHAPE). *J. Am. Chem. Soc.* **127**, 4223–31 (2005).
18. Yang, M. *et al.* Intact RNA structurome reveals mRNA structure-mediated regulation of miRNA cleavage *in vivo*. *Nucleic Acids Res.* **48**, 8767–8781 (2020).
19. Smola, M. J. & Weeks, K. M. In-cell RNA structure probing with SHAPE-MaP. *Nat. Protoc.* **13**, 1181–1195 (2018).
20. Yang, X. *et al.* RNA G-quadruplex structures exist and function *in vivo* in plants. *Genome Biol.* **21**, 226 (2020).
21. Liu, Z. *et al.* *In vivo* nuclear RNA structurome reveals RNA-structure regulation of mRNA processing in plants. *Genome Biol.* **22**, 11 (2021).
22. Tian, S., Cordero, P., Kladwang, W. & Das, R. High-throughput mutate-map-rescue evaluates SHAPE- directed RNA structure and uncovers excited states. doi:10.1261/rna.044321.114
23. Y, D. *et al.* *In vivo* genome-wide profiling of RNA secondary structure reveals novel regulatory features. *Nature* **505**, 696–700 (2014).
24. Li, P., Tao, Z. & Dean, C. Phenotypic evolution through variation in splicing of the noncoding RNA *COOLAIR*. *Genes Dev.* **29**, 1–6 (2015).
25. Zhu, P., Lister, C. & Dean, C. Cold-induced *Arabidopsis* FRIGIDA nuclear condensates for *FLC* repression. *Nature* **599**, 657–661 (2021).

Reviewer Reports on the First Revision:

Referees' comments:

Referee #1:

The authors have revised the manuscript to improve the single molecule RNA structural analysis and functional data of COOLAIR conformation on plant flowering time. The functional data on RNA structural compensatory mutation supports the authors' model and is satisfactory. However, the new single molecule RNA structure analysis makes several claims that are not fully supported by data.

1. The authors agree with prior critiques of the smStructure-seq probing analysis (DREEM) and now develop a new method, named Davinci, to analyze data. The validation of this approach appears incomplete. In the validation case of Rev response element (RRE), the authors report that Davinci identified a rare conformation of the RRE (conformation 3). However, the published literature (Legiewicz et al., PNAS, 2008, ref 6) indicates that conformation 3 is not detected in wild-type RRE but only in mutant RRE with two G to A mutations.

2. Similarly, the validation case for pri-miRNA conformations in chr2 mutant, the authors report new conformational subspecies that they identified. But the existence of such species was not previously known or validated by orthogonal means, raising the question how we know results from a new method are correct. Analysis of RNAs with well-known conformational dynamics, e.g. riboswitches, would be a more fruitful to prove out the new analysis framework of smStructure-seq.

3. In sum, the authors have generated new alleles of COOLAIR RNA that suggest that lncRNA isoforms and secondary structures alter lncRNA function in flower timing. The caveats above about the RNA structure probing data renders the molecular interpretation of the genetic data somewhat uncertain.

Referee #2:

In this revision the authors have answered the comments I made in the previous review, notably adding additional data on the COOLAIR structures in the mutant line and improving the presentation to make the manuscript easier to follow.

Referee #3:

The authors have largely addressed my concerns.

Author Rebuttals to First Revision:

Referee #1:

The authors have revised the manuscript to improve the single molecule RNA structural analysis and functional data of COOLAIR conformation on plant flowering time. The functional data on RNA structural compensatory mutation supports the authors' model and is satisfactory. However, the new single molecule RNA structure analysis makes several claims that are not fully supported by data.

1. The authors agree with prior critiques of the smStructure-seq probing analysis (DREEM) and now develop a new method, named Davinci, to analyze data. The validation of this approach appears incomplete. In the validation case of Rev response element (RRE), the authors report that Davinci identified a rare conformation of the RRE (conformation 3). However, the published literature (Legiewicz et al., PNAS, 2008, ref 6) indicates that conformation 3 is not detected in wild-type RRE but only in mutant RRE with two G to A mutations.

Response

We have undertaken analyses to show that the RRE wild-type sequence has the potential to form conformation 3. We have also undertaken experiments to show that this conformation becomes the dominant one when the mutations are included.

In silico RNA structure ensembles of wild-type RRE and mutant RRE61 were generated by Boltzmann sampling (10,000 times) using RNAfold¹. Three structural clusters (Rebuttal Fig. 1a) were found with conformation 3 being the least abundant (1%). With the mutant RRE61², conformation 3 increased to 95.6% (Rebuttal Fig. 1b). Thus, the wild-type RRE sequence has the potential to fold into the rare conformation 3 with the mutation converting it to the dominant conformation.

To experimentally confirm the conformational change caused by the mutation in RRE61, we folded the RRE61 RNA and probed its structure using the same conditions as in the previous study³. We then performed our DaVinci analysis on RRE61. Indeed, our experimental results showed that conformation 3 increased to 81% (Rebuttal Fig. 1d) from 2% in RRE³ (Rebuttal Fig. 1c). DaVinci directly measures the percentage of clusters by counting each single RNA structure derived from the probing data and this contrasts with *in silico* RNA structure ensemble analysis, where Boltzmann sampling measures the percentage using a function of free energy. Thus, DaVinci analysis can estimate the genuine proportions and distributions of each conformation cluster.

Therefore, our analysis further confirms that mutation or single nucleotide polymorphisms (SNP) alter the RNA structural ensemble and change the proportions of the different conformations^{4,5}. We have included this information in Extended Data Fig. 2 and the corresponding text in the Supplementary Discussion in line (77-95).

Rebuttal Fig. 1 (Extended Data Fig. 2b-e). *In silico* structural ensemble of RRE wild-type (a) and RRE61 mutant (b). Structures were predicted using Boltzmann suboptimal sampling. c, The RNA structural ensemble of *in vitro* RRE RNA by DaVinci with the guidance of DMS reactivity (from Phillip J. Tomezsko, et.al. 2020³). d, The RNA structural ensemble of *in vitro* RRE61 by Davinci with the guidance of *in vitro* DMS reactivity from our probing experiment. The altered stem-loop structures are labelled as III, IV and V as in^{2,6}.

2. Similarly, the validation case for *pri-miRNA* conformations in *chr2* mutant, the authors report new conformational subspecies that they identified. But the existence of such species was not previously known or validated by orthogonal means, raising the question how we know results from a new method are correct. Analysis of RNAs with well-known conformational dynamics, e.g. riboswitches, would be a more fruitful to prove out the new analysis framework of smStructure-seq.

Response

Following the reviewer's suggestion, we not only validated the efficacy of DaVinci on analysing the structure of a thermometer, *Escherichia coli* *cspA* 5' untranslated region (UTR) with a published dataset, but also experimentally tested DaVinci on a TPP riboswitch. All the results show that DaVinci can sensitively and accurately capture the RNA conformations.

This *cspA* 5' UTR functions as an RNA thermometer since it can switch states between translationally repressed conformations (conformation 3 and 4) at 37 °C and translationally competent conformations (conformation 1 and 2) at 10 °C^{7,8}. DaVinci results showed that the translationally competent conformations (conformation 1 and 2) increased from 23% to 67% upon transfer from 37 °C to 10 °C (Rebuttal Fig. 2). These two conformations have been previously detected after the cold treatments⁸. DaVinci also identified an extra conformation 3, which is very similar to the major conformation 4⁸ at

37 °C. Compared with the conformation 4, conformation 3 loses a short stem loop, further indicating that DaVinci is sensitive to detect less abundant RNA structural conformations.

Rebuttal Fig. 2 (Extended Data Fig. 3). Davinci-determined RNA structure conformation space for *cspA* RNA folded and probed at 37 °C (a) or 10 °C (b) (from Zhang et.al., 2018⁸). The red rectangle is the start codon.

To further experimentally test our DaVinci, we took advantage of a TPP riboswitch, a typical RNA molecule which can fold into alternative structures depending on the presence of the TPP ligand^{9–11}. We performed the RNA structure probing experiments on *in vitro* folded TPP riboswitch RNAs (*TenA* gene in *B. subtilis*) in the absence or presence of 1 μM TPP ligand. After the treatment of the SHAPE chemicals (NAI), we merged the NAI-modified RNA samples (TPP-treated and non TPP-treated RNAs) with a ratio of 20:80(v/v) or 50:50(v/v) and conducted the library constructions, respectively. We then performed our DaVinci analysis on the sequencing data. We found that DaVinci closely reflects the different ratio of the two alternative conformations (Conformation 1 is related to the TPP-treated conformation and Conformation 2 is related to the non-TPP treated one, Rebuttal Fig. 3) with the ratios of 29:71 or 40:60. Overall, DaVinci accurately detects RNA structural conformations. The slight difference between the expected ratios and the DaVinci-derived ratios is likely to reflect the equilibrium of the conformations during the RNA structure probing in the solutions.

We have included this information in Extended Data Fig. 2, Extended Data Fig. 3 and the corresponding text in the Supplementary Discussion in lines (97-120).

We had used *pri-miRNA319* as an example to indicate that the RNA structure change of *pri-miRNA319* based on the RNA populations might be derived from the re-distribution of diverse RNA structural conformations. Although there are no reports of these structures by other orthogonal methods, there are some indirect pieces of evidence which may support the existence of these the conformations. For example, the conformation 1 is very similar to the miRNA319b isoform¹²; the conformation 2 is very similar to the miRNA319c¹²; conformation 3 is very similar to the isoform of miRNA319a¹². To avoid confusing the readers, we removed these results from the text in the Supplementary Discussion.

Rebuttal Fig. 3 (Extended Data Fig. 2 f-h). **a**, the DaVinci-determined RNA structure conformation space for TenA RNA folded with and without $1 \mu\text{M}$ TPP ligands. The folded RNAs were probed *in vitro* and pooled at the ratio of 20 (with TPP):80 (without TPP). **b**, Similar to **(a)** with the pooling ratio of 50 (with TPP):50 (without TPP). **c**, Proportions for each cluster detected by DaVinci. The colour corresponding to **a** and **c**.

3. In sum, the authors have generated new alleles of COOLAIR RNA that suggest that lncRNA isoforms and secondary structures alter lncRNA function in flower timing. The caveats above about the RNA structure probing data renders the molecular interpretation of the genetic data somewhat uncertain.

Our results strongly support that DaVinci can provide sensitive and accurate detection of RNA structural conformations from experimental data. Furthermore, DaVinci provides a new framework based on single molecule RNA structure probing to investigate RNA structural conformation. It also avoids the over-clustering of reactivity into two extreme mutation profiles (detailed in Supplementary Discussion). DaVinci's experimentally derived framework is also independent of thermodynamic parameters. It allows direct measurement of the percentage of each conformation cluster, and thus avoids the bias resulted from Boltzmann sampling which is relying on the function of free energy. Therefore, our smStructure-seq method equipped with DaVinci RNA analysis will facilitate the study of RNA structural conformation *in vivo*. More importantly, our study on *in vivo* COOLAIR structure will be an example for other lncRNA studies and we have edited the manuscript to strengthen the biological significance.

References

1. Hofacker, I. *et al.* Fast Folding and Comparison of RNA Secondary Structures. *Monatsh Chem (Chem Mon)*. **125**, 167–188 (1994).
2. Legiewicz, M. *et al.* Resistance to RevM10 inhibition reflects a conformational switch in the HIV-1 Rev response element. *Proc. Natl. Acad. Sci. U. S. A.* **105**, 14365–14370 (2008).
3. Tomesko, P. J. *et al.* Determination of RNA structural diversity and its role in HIV-1 RNA splicing. *Nature* **582**, 438–442 (2020).
4. Halvorsen, M., Martin, J. S., Broadaway, S. & Laederach, A. Disease-associated mutations that alter the RNA structural ensemble. *PLoS Genet.* **6**, e1001074 (2010).
5. Linnstaedt, S. D. *et al.* A Functional riboSNitch in the 3' Untranslated Region of FKBP5 Alters MicroRNA-320a Binding Efficiency and Mediates Vulnerability to Chronic Post-Traumatic Pain. *J. Neurosci.* **38**, 8407–8420 (2018).
6. Sherpa, C., Rausch, J. W., Le Grice, S. F. J., Hammarskjold, M. L. & Rekosh, D. The HIV-1 Rev response element (RRE) adopts alternative conformations that promote different rates of virus replication. *Nucleic Acids Res.* **43**, 4676–4686 (2015).
7. Giuliodori, A. M. *et al.* The cspA mRNA Is a Thermosensor that Modulates Translation of the Cold-Shock Protein CspA. *Mol. Cell* **37**, 21–33 (2010).

8. Zhang, Y. *et al.* A Stress Response that Monitors and Regulates mRNA Structure Is Central to Cold Shock Adaptation. *Mol. Cell* **70**, 274–286.e7 (2018).
9. Edwards, T. E. & Ferré-D'Amaré, A. R. Crystal Structures of the Thi-Box Riboswitch Bound to Thiamine Pyrophosphate Analogs Reveal Adaptive RNA-Small Molecule Recognition. *Structure* **14**, 1459–1468 (2006).
10. Quarta, G., Kim, N., Izzo, J. A. & Schlick, T. Analysis of Riboswitch Structure and Function by an Energy Landscape Framework. *J. Mol. Biol.* **393**, 993–1003 (2009).
11. Manzourolajdad, A. & Arnold, J. Secondary structural entropy in RNA switch (Riboswitch) identification. *BMC Bioinformatics* **16**, (2015).
12. Sobkowiak, L., Karlowski, W., Jarmolowski, A. & Szweykowska-Kulinska, Z. Non-canonical processing of *Arabidopsis* pri-miR319a/b/c generates additional microRNAs to target one RAP2.12 mRNA isoform. *Front. Plant Sci.* **3**, 1–12 (2012).

Reviewer Reports on the Second Revision:

Referees' comments:

Referee #1 (Remarks to the Author):

The authors have satisfactorily addressed my concerns. I support publication of this work.